# Uncertainty-Weighted Fusion of RGB and Synthetic Motion Cues for Video Anomaly Detection

## Abstract

Most existing video anomaly detectors rely solely on RGB frames, which lack the temporal resolution needed to capture abrupt or transient motion cues—key indicators of anomalous events. To address this, we introduce a robust framework for video anomaly detection that effectively fuses complementary RGB and synthetic motion cues. Our approach, Uncertainty-Weighted Image-Event Fusion (IEF-VAD), addresses the modality imbalance inherent in such data by using a principled, uncertainty-aware process. The system (i) models the high variance and heavy-tailed noise of synthetic cues with a Student's t likelihood; (ii) derives value-level inverse-variance weights via a Laplace approximation to prevent the dominant image modality from suppressing motion-centric signals; and (iii) iteratively refines the fused latent state to remove residual cross-modal noise. This uncertainty-driven fusion consistently outperforms conventional fusion methods like cross-attention and gating, which are prone to modality dominance. Without any dedicated event sensor or frame-level labels, IEF-VAD sets a new state of the art across multiple real-world anomaly detection benchmarks, demonstrating robust performance even under modality-specific degradation. These findings highlight the utility of extracting and integrating these complementary motion cues for accurate and robust video understanding across diverse applications.

## 1 Introduction

Recent advances in deep learning have led to significant progress in multimodal data analysis Xu et al. (2023); Liang et al. (2024); Girdhar et al. (2023); Radford et al. (2021); Li et al. (2024); Guo et al. (2019); Akbari et al. (2021), enabling systems to effectively integrate diverse sensory inputs for complex tasks such as video anomaly detection Flaborea et al. (2023); Zhang et al. (2024); Tang et al. (2024); Ji et al. (2020); Feng et al. (2023). Traditionally, research in this area has predominantly focused on leveraging static image information, which provides rich spatial context. However, dynamic motion cues—capturing subtle temporal changes, abrupt transitions, and transient patterns—remain largely underexplored. Real event data inherently provides microsecond resolution and polarity information, making it well suited to capture fleeting anomalies that static images often miss Gallego et al. (2020); Rebecq et al. (2019); Wang et al. (2022); Chakravarthi et al. (2024). Yet, the scarcity of real-world event datasets and sensor-specific variability has limited their use in anomaly detection. In this work, we derive *synthetic motion cues* from RGB videos via simple inter-frame differencing and thresholding. Although lacking the precise temporal fidelity and polarity of true event streams, these cues retain the key property of events—sparse, background-suppressed, motion-centric representations. We therefore refer to them as *event-like* proxies: not replacements for real sensors, but scalable motion-salient cues effectively leveraged through our uncertainty-aware fusion framework.

The challenge of effectively combining homogeneous data sources—static RGB information and its derived synthetic motion cues—under conditions of uncertainty remains a critical open problem. Recent transformer-based fusion methods for video anomaly detection Feng et al. (2021b); Wu et al. (2020); Zhou et al. (2019); Zhu et al. (2013); Ding et al. (2025); Ghadiya et al. (2024) still focus on the rich spatial details in images and, as a result, under-utilize the transient yet crucial temporal cues in motion representations. Because both streams originate from the same modality, their homogene-

ity makes naive fusion especially prone to *modality dominance*, where the dense spatial features suppress complementary motion-centric information Park et al. (2024); Tsai et al. (2019); Liu et al. (2021a). These limitations highlight the need for a fusion strategy that can explicitly account for uncertainty and rebalance contributions across homogeneous yet imbalanced feature spaces.

To address these challenges, we propose an approach that explicitly fuses RGB features with synthetic motion cues through an uncertainty-aware framework. Our uncertainty-weighted Image and Event proxy Fusion framework for Video Anomaly Detection (*IEF-VAD*) balances the two inputs by assigning inverse-variance weights derived from Bayesian uncertainty estimates. By modelling each stream's latent features alongside their predictive variance, IEF-VAD down-weights less reliable signals and prevents the RGB stream, richer in spatial content but often dominant, from overshadowing the temporally informative motion cues. Concretely, we capture the heavy-tailed noise of motion cues with a Student's t likelihood and obtain Gaussian approximation via Laplace method Malmström et al. (2023); Zhu et al. (2013); Daxberger et al. (2021); Wu et al. (2021), while drawing on established Bayesian techniques for uncertainty estimation Zhu et al. (2013); Subedar et al. (2018); Kendall & Gal (2017); Ober et al. (2021); Gal & Ghahramani (2016). The resulting uncertainty-weighted fusion dynamically modulates each input's contribution, allowing high-resolution temporal cues to complement, rather than be suppressed by, the detailed spatial context of images.

To address this imbalance at a finer granularity, we model uncertainty at the feature level by assigning each latent dimension its own variance estimate. Concretely, we assume a Student's-$t$ likelihood for each value to capture heavy-tailed noise, particularly prevalent in motion cues, and approximate it with a Gaussian via a Laplace expansion. This yields effective per-dimension variances that are converted into precision weights, enabling value-level inverse-variance fusion. In this way, unreliable features are down-weighted, and complementary motion cues retain influence even when co-present with dominant RGB content.

Furthermore, our framework incorporates a sequential update mechanism and iterative refinement process. The sequential update merges each new image–motion cue observation with the prior fused state via inverse-variance (Kalman-gain) weights, closely mirroring the Kalman filter update step Welch & Bishop (1995); Haarnoja et al. (2016). Iterative refinement then targets the fine-grained residual errors that the fusion step cannot fully resolve—such as feature-level mismatches, modality-specific noise, and minor scale imbalances that arise when combining two streams with different noise profiles. By repeatedly estimating and subtracting these residuals, the refinement network progressively denoises and re-balances the fused representation, yielding a cohesive latent state.

Our contributions can be summarized as follows:

- **Precision-Weighted Fusion Mechanism:** We introduce an uncertainty-aware fusion strategy that assigns inverse-variance (precision) weights to image and motion-cue representations, inspired by Bayesian inference and Kalman filtering. This approach adaptively balances contributions by down-weighting unreliable signals, preventing dominant RGB features from suppressing motion-centric cues.

- **Practical Integration of Synthetic Motion Cues:** We demonstrate that motion cues derived from conventional RGB videos can be effectively integrated into anomaly detection pipelines. This provides a scalable alternative to real event sensors and broadens the applicability of motion-aware fusion across diverse video datasets.

- **Empirical Validation on Real-World Datasets:** Our framework achieves state-of-the-art results on four benchmark datasets—UCF-Crime (88.67% AUC), XD-Violence (87.63% AP), ShanghaiTech (97.98% AUC), and MSAD (92.90% AUC). Beyond aggregate performance, our analysis shows that motion cues are particularly decisive in motion-dominant anomaly classes (e.g., *Assault*, *Fighting*, *Shoplifting*), where they significantly boost detection accuracy compared to RGB-only or conventional fusion methods.

## 2 RELATED WORK

**Adaptive Multimodal Fusion.** Multimodal fusion integrates complementary signals from vision, audio, and language Xu et al. (2023); Zong et al. (2023); Huang et al. (2022). Early work relied on feature concatenation Ngiam et al. (2011), while recent methods employ transformers Tsai et al.

(2019); Akbari et al. (2021); Girdhar et al. (2022); Li et al. (2021) and contrastive pretraining Radford et al. (2021); Girdhar et al. (2023); Dai et al. (2022). Attention mechanisms improve alignment but risk modality dominance Liang et al. (2024); Xu et al. (2023). Recent studies use LLMs to modulate modality usage dynamically Shen et al. (2023); Zhao et al. (2023); Gong et al. (2023); Li et al. (2023); Driess et al. (2023). Complementary to these, we emphasize *uncertainty* as an explicit control signal for robust fusion under imbalance and noise.

**Event Cues for Video Anomaly Detection.** Event-based sensors provide sparse, high-temporal-resolution signals Gallego et al. (2020); Zheng et al. (2023); Chakravarthi et al. (2024), applied in recognition Shiba et al. (2022); Yang et al. (2023); Luo et al. (2023) and modality alignment with CLIP Wu et al. (2023); Zhou et al. (2024); Jeong et al. (2024). Meanwhile, video anomaly detection has advanced through MIL-based weak supervision Sultani et al. (2018); Wu et al. (2020); Zhong et al. (2022); Chen et al. (2024), yet remains image-centric Liu et al. (2021b); Georgescu et al. (2021). Recent work explores multimodal cues, including text via LLMs Alayrac et al. (2022); Driess et al. (2023); Zhao et al. (2023). Our approach leverages *synthetic motion cues*—event-like proxies derived from RGB videos Rebecq et al. (2019); Zhu et al. (2018); Astrid et al. (2021); Jeong et al. (2024)—to capture motion-salient dynamics without dedicated event sensors.

**Uncertainty and Bayesian Fusion.** Uncertainty estimation adapts model confidence under noise and distributional shift Kendall & Gal (2017); Gal & Ghahramani (2016); Ovadia & et al. (2019); Subedar et al. (2018). Bayesian methods such as dropout inference and inverse-variance weighting Gal & Ghahramani (2016); Subedar et al. (2018); Daxberger et al. (2021) improve robustness. We extend this line by combining Laplace approximation with a Student's t noise model Malmström et al. (2023); Wu et al. (2021), enabling principled fusion of image and motion-cue features under heavy-tailed uncertainty.

## 3 Uncertainty-Weighted Fusion for Multimodal Learning

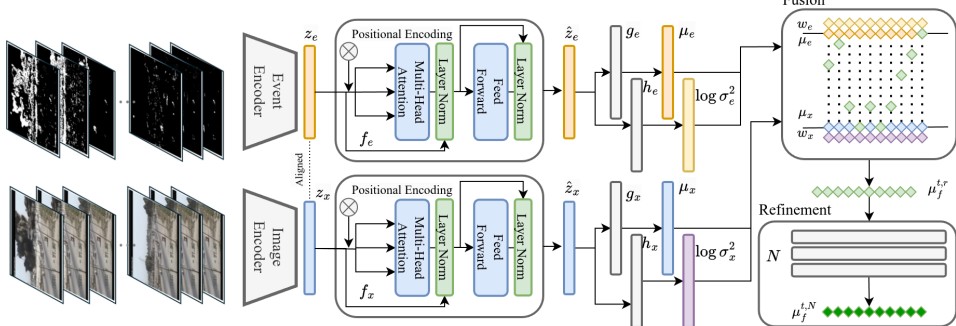

Figure 1: **Overview of IEF-VAD framework.** Each video frame and its corresponding synthetic motion cue are processed by CLIP encoders to obtain feature embeddings $z_m$. These are further encoded by modality-specific transformers $f_m$ to produce $\hat{z}_m$, which are then passed through projection heads $g_m$ and $h_m$ to estimate $\mu_m$ and $\sigma_m$. The estimated $\sigma_m$ is used to compute the uncertainty-aware fusion weight $w_m$, which is applied to obtain the initial fused representation $\mu_f^{t,0}$. This representation is then refined over $N$ iterative steps through a refinement network to produce the final output $\mu_f^{t,N}$.

In this work, we extract image embeddings ($z_x$) and motion embeddings ($z_e$) separately from videos, where $z_e$ denotes an *event-like proxy* (synthetic motion cue) derived from RGB frames. We assume that both streams observe the same underlying scene and share a common spatial structure while exhibiting complementary expressive and modality-specific features. $z_x$ aggregates complex attributes such as color, background, and spatial details, whereas $z_e$ encapsulates transient changes and temporal dynamics. To effectively combine these homogeneous yet imbalanced representations, we propose a fusion strategy that integrates Bayesian uncertainty estimation with Kalman filter update principles, where uncertainty quantifies the reliability of each value-level feature. Specifically, we estimate latent features and their associated uncertainties for each stream, and subsequently fuse them through a refinement network designed to iteratively correct residual noise, misalignments, and modality-specific artifacts, yielding a robust final representation $\mu_f$.

## 3.1 SYNTHETIC MOTION CUE

We derive a *synthetic motion cue* directly from RGB frames to leverage temporal dynamics. Given consecutive grayscale frames $I_t$ and $I_{t+1}$, we compute the absolute difference $\Delta_t(x,y) = |I_{t+1}(x,y) - I_t(x,y)|$. Pixels are activated when the difference exceeds a threshold, $E_t(x,y) = \mathbb{1}[\,|\Delta_t(x,y)| > \tau\,]$, and aggregated over a short temporal window $T$ as $\hat{E}(x,y) = \frac{1}{T}\sum_{t=1}^{T} E_t(x,y)$.

This event-like representation suppresses static background while isolating localized motion, providing a lightweight proxy for event data without specialized sensors. Although derived from RGB, it complements image features by highlighting transient cues that are often diluted in appearance-only representations, and empirical results confirm its robustness and contribution to anomaly detection. Further implementation details and extended empirical analyses can be found in Appendix A.

## 3.2 MODALITY DESIGN AND UNCERTAINTY ESTIMATION

We design each modality as a noisy observation influenced by both shared scene content and modality-specific factors. $z_m = \mu_m + \delta_m$, $\delta_m \sim t_\nu(0, \Sigma_m)$, $m \in \{x, e\}$, where $\mu_m \in \mathbb{R}^d$ is the central estimate and $\Sigma_m = \mathrm{diag}(\sigma_m^2)$ encodes value-level uncertainty. We choose the heavy-tailed Student's-$t$ distribution because a purely Gaussian model underestimates uncertainty in the presence of outliers; its thicker tails make the fusion rule more conservative when the input is degraded (see Appendix C).

Given embedding sequences $z_m \in \mathbb{R}^{B \times T \times D}$, we transpose them to $(T, B, D)$, pass each through a modality-specific transformer $f_m$, and transpose back, yielding $\hat{z}_m = (f_m(z_m^\top))^\top$.

Layer normalization aligns scales across modalities. This ensures that the downstream uncertainty estimation operates on comparable feature scales, improving numerical stability when computing variances. Linear projection heads $g_m, h_m$ then predict the posterior mean and log-variance:

$$\mu_m = g_m(\hat{z}_m), \quad \log \sigma_m^2 = h_m(\hat{z}_m). \tag{1}$$

Predicting $\log \sigma_m^2$ guarantees positivity and numerical stability. Conceptually, $z_m$ is sampled from the posterior $t_\nu(\mu_m, \Sigma_m)$, where $\mu_m$ represents the fused central tendency and $\sigma_m^2$ quantifies epistemic uncertainty. These uncertainty estimates drive the Kalman-style update and the subsequent refinement loop, enabling dynamic, reliability-aware fusion that adapts to modality quality in real time.

For computational tractability, we adopt a diagonal covariance structure, treating each feature dimension as conditionally independent. This avoids the $\mathcal{O}(D^3)$ cost of full-covariance modeling while enabling efficient per-dimension uncertainty estimation. A detailed discussion of this design choice, along with empirical justification, is provided in Appendix D.

## 3.3 INVERSE VARIANCE CALCULATION

In this stage, our goal is to quantify the confidence in each modality's predictions by computing the inverse variance. However, direct computation with the Student's t distribution is intractable for variance inversion and KL divergence, so we approximate it with a Gaussian via Laplace expansion (see Appendix B). Recall that our model predicts the log-variance values (Eq. 1), which are then exponentiated to recover the variance. In the case of a Gaussian noise model, this is given by $\sigma_m^2 = \exp\left(\log \sigma_m^2\right)$. For the Student's t noise model, we first apply the Laplace approximation to obtain the effective variance (see Appendix E). The Student's t probability density function (up to a normalization constant) is given by Eq. 2, where $\sigma^2$ is the variance parameter for the underlying Gaussian scale, and $\nu$ is the degree of freedom.

$$p(\delta) \propto \left(1 + \frac{\delta^2}{\nu\sigma^2}\right)^{-\frac{\nu+1}{2}}, \quad \log p(\delta) = -\frac{\nu+1}{2} \log \left(1 + \frac{\delta^2}{\nu\sigma^2}\right) + C. \tag{2}$$

Since the mode of the distribution occurs at $\delta = 0$, we perform a second-order Taylor expansion of the logarithm around $\delta = 0$. For small $x$, recall that $\log(1 + x) \approx x$ for $|x| \ll 1$. In our case, set

$x = \delta^2/(\nu\sigma^2)$. Then, for small $\delta$, and log-density we have $\log\left(1 + \frac{\delta^2}{\nu\sigma^2}\right) \approx \frac{\delta^2}{\nu\sigma^2}$, $\log p(\delta) \approx$ $-\frac{\nu+1}{2}\frac{\delta^2}{\nu\sigma^2} + C = -\frac{\nu+1}{2\nu\sigma^2}\delta^2 + C$. The Laplace approximation approximates a probability density near its mode by a Gaussian distribution. The log-density of a Gaussian with variance $\tilde{\sigma}^2$ is given by $\log p_G(\delta) = -\frac{1}{2\tilde{\sigma}^2}\delta^2 + C$. By matching the quadratic terms in the Taylor expansion, we set $\frac{1}{2\tilde{\sigma}^2} = \frac{\nu+1}{2\nu\sigma^2}$. This immediately implies $\tilde{\sigma}^2 = \frac{\nu}{\nu+1}\sigma^2$. Taking the logarithm of both sides gives

$$\log\tilde{\sigma}^2 = \log\sigma^2 + \log\left(\frac{\nu}{\nu+1}\right). \tag{3}$$

This derivation shows that, under the Laplace approximation, the effective variance used in downstream computations is scaled by the factor $\frac{\nu}{\nu+1}$, reflecting the heavy-tailed nature of the Student's t distribution. This effective variance is then used in place of the original variance $\sigma^2$ when computing inverse variance weights and other related measures, ensuring that the fusion process properly accounts for the increased uncertainty due to heavy-tailed noise.

The variance (or effective variance in the Student's t case) represents the uncertainty associated with the prediction; lower values indicate higher confidence. To leverage this notion of confidence in a manner consistent with Bayesian principles, we compute the inverse variance as a measure of precision. Specifically, for the Student's t model, we use the effective variance:

$$w_m = \frac{1}{\tilde{\sigma}_m^2 + \epsilon}. \tag{4}$$

where $\epsilon$ is a small positive constant added for numerical stability, ensuring that we do not encounter division by zero. These weights, $w_x$ and $w_e$, essentially serve as confidence scores—modalities with lower uncertainty (i.e., lower $\sigma^2$ or $\tilde{\sigma}^2$) yield higher weights and, consequently, contribute more significantly in subsequent fusion steps. This approach is theoretically grounded in Bayesian inference (detailed in Appendix F), where the inverse variance (or precision) is used to weight the contributions of different measurements according to their reliability. As a result, by explicitly modeling and incorporating the inverse variance—or the effective inverse variance under the Student's t assumption—the fusion process becomes more robust, effectively balancing the contributions of each modality based on their estimated uncertainties.

### 3.4 Uncertainty-Weighted Fusion

We compute the fused representation $\mu_f \in \mathbb{R}^{B \times T \times D}$ by taking a weighted average of the modality-specific means according to their computed confidence scores Eq. 4. $\mu_f = \frac{w_x\mu_x + w_e\mu_e}{w_x + w_e}$ In the case of the Student's t distribution, the inverse variance weights are computed based on the effective variances obtained via a Laplace approximation around the mode of the distribution. Specifically, by applying the logarithm to the effective variance as derived in Eq. 3, and then exponentiating, we have $\tilde{\sigma}_m^2 = \exp\left(\log\sigma_m^2 + \log\left(\frac{\nu}{\nu+1}\right)\right)$ Thus, in both cases, modalities with lower uncertainty (i.e., lower variance or effective variance) yield higher precision scores and contribute more significantly in the fusion process.

The rationale behind this formulation is twofold. First, by weighting each modality's mean by its (effective) inverse variance, we ensure that modalities with higher confidence have a greater impact on the final fused representation. This is a direct application of Bayesian fusion principles, where the posterior estimate of a latent variable is a precision-weighted average of the individual estimates. Second, this fusion strategy closely mirrors the update step in the Kalman filter—a well-established method for sequential data fusion—where the Kalman gain, derived from the inverse variances, dictates the contribution of each measurement in updating the state estimate, as further detailed in Appendix F.

### 3.5 Time Step Sequential Update

Building upon our uncertainty-weighted fusion framework (Figure 1), we extend the method to handle the time steps ($T$) by incorporating temporal dependencies in a sequential update process. The intuition is analogous to the recursive estimation in Kalman filtering, except that here we account for the heavy-tailed nature of the Student's t noise via a Laplace approximation.

Under the Student's t model, the predicted variance is corrected to obtain an effective variance. Specifically, if the predicted log-variance is $\log \sigma_t^2$ at time $t$, the effective variance and state variance at the previous time step are given by

$$\tilde{\sigma}_t^2 = \exp\left(\log \sigma_t^2 + \log\left(\frac{\nu}{\nu+1}\right)\right) \tag{5}$$

At the initial time step ($t = 0$), we set the state and its effective uncertainty directly from the first input: $\mu_f^0 = \mu^0$, $\tilde{\sigma}_{f,0}^2 = \tilde{\sigma}_0^2$. For each subsequent time step ($t \geq 1$), we update the state estimate by fusing the previous state with the current input. To do so, we compute the inverse effective variance weights. The weight for the previous state is given by $w_f^{t-1} = 1/(\tilde{\sigma}_{f,t-1}^2 + \epsilon)$ and the weight for the current input is $w^t = 1/(\tilde{\sigma}_t^2 + \epsilon)$, where $\epsilon$ is a small constant for numerical stability. The updated state and its effective uncertainty are computed as:

$$\mu_f^t = \frac{w_f^{t-1}\mu_f^{t-1} + w^t\mu^t}{w_f^{t-1} + w^t}, \quad \tilde{\sigma}_{f,t}^2 = \frac{1}{w_f^{t-1} + w^t}, \quad t \geq 1. \tag{6}$$

This sequential update process naturally extends the static fusion methodology by incorporating temporal continuity, using effective variances that account for the heavy-tailed nature of the Student's t noise. Just as in the static case—where each modality is weighted according to its (effective) precision—the sequential update fuses the previous state with new observations based on their relative effective uncertainties. This approach enhances the robustness and consistency of the state estimates over time, effectively capturing both current observations and historical context.

## 3.6 ITERATIVE REFINEMENT OF FUSED STATE

After sequential fusion via uncertainty-weighted averaging, residual noise and discrepancies between modalities inevitably remain in the fused latent state due to their distinct noise profiles. To address these fusion-induced residuals, we introduce an iterative refinement procedure inspired by denoising methods Ho et al. (2020).

Formally, starting from an initial fused state $\mu_f^0$, the refinement network $F(\cdot)$ predicts the residual error $\Delta\mu_f^{t,r} = F(\mu_f^{t,r})$ at iteration $r$ ($r = 0, 1, \ldots, N$). The state is updated with an attenuation parameter $\lambda_r \in (0, 1)$:

$$\mu_f^{t,r+1} = \mu_f^{t,r} - \lambda_r \Delta\mu_f^{t,r}.$$

This parameter prevents over-correction and ensures stable convergence. As refinement proceeds, residual errors diminish, aligning the fused state closer to the true latent representation. Thus, iterative refinement provides a principled mechanism to denoise and enhance multimodal fusion, yielding more reliable embeddings (Appendix G).

## 3.7 VIDEO ANOMALY DETECTION WITH LOSS FUNCTIONS

**Classification Loss ($\mathcal{L}_{\textbf{cls}}$):** Given the refined fused representation $\mu_f^{t,N} \in \mathbb{R}^{B \times T \times D}$, we apply a lightweight head $H(\cdot)$ to produce logits $\hat{y} = H(\mu_f^{t,N}) \in \mathbb{R}^{B \times T \times 1}$. After sigmoid activation, anomaly probabilities are trained with binary cross-entropy:

$$\mathcal{L}_{\text{cls}} = -\frac{1}{BT}\sum_{b=1}^{B}\sum_{t=1}^{T}[y_{b,t}\log(\hat{y}_{b,t}) + (1 - y_{b,t})\log(1 - \hat{y}_{b,t})].$$

To allow weak supervision without frame-level labels, we follow Sultani et al. (2018) and divide each sequence into non-overlapping 16-frame segments. Let length be the number of valid steps, then $k = \lfloor\frac{\text{length}}{16}\rfloor + 1$, and segment-level predictions are obtained by averaging frame scores Ilse et al. (2018); Sultani et al. (2018).

**KL Divergence Loss ($\mathcal{L}_{\textbf{KL}}$):** Each modality's noise follows $\delta_m \sim t_\nu(0, \sigma_m^2)$. Since the KL divergence with a standard normal prior is intractable, we approximate the Student's $t$ by a Gaussian

$\mathcal{N}(\mu_m, \tilde{\sigma}_m^2)$ using the effective variance $\tilde{\sigma}_m^2$ from Laplace approximation (Eq. 5). The closed-form KL divergence is:

$$\mathcal{L}_{\text{KL}} = KL\big(\mathcal{N}(\mu_m, \tilde{\sigma}_m^2) \,\|\, \mathcal{N}(0, I)\big) = \tfrac{1}{2}\big(\tilde{\sigma}_m^2 + \mu_m^2 - 1 - \log \tilde{\sigma}_m^2\big).$$

This regularization keeps the latent distribution close to the prior, enhancing robustness under heavy-tailed noise.

**Regularization Loss ($\mathcal{L}_{\text{reg}}$):** To align latent representations $\mu_x$ and $\mu_e$ in both direction and magnitude, we define

$$\mathcal{L}_{\text{reg}} = \lambda_1(1 - \cos(\mu_x, \mu_e)) + \lambda_2\big|\|\mu_x\| - \|\mu_e\|\big|,$$

where $\lambda_1, \lambda_2$ balance the two terms. The cosine term enforces semantic alignment, while the norm term prevents scale imbalance, together promoting a coherent multimodal space.

**Overall Loss:** The final loss function is a sum of the classifier loss, the KL divergence losses for both modalities, and the regularization loss:

$$\mathcal{L} = \mathcal{L}_{\text{cls}} + \sum_m \mathcal{L}_{\text{KL}}^m + \mathcal{L}_{\text{reg}} \qquad (7)$$

This composite loss enforces accurate predictions while encouraging robust, well-regularized representations. By using the effective variance from Laplace's approximation, we retain a closed-form KL divergence, combining the robustness of Student's t with the efficiency of Gaussian methods. An ablation of loss components is given in Appendix J.

## 4 EXPERIMENT RESULTS

We evaluate our method on four public video anomaly detection datasets—UCF-Crime Sultani et al. (2018), XD-Violence Wu et al. (2020), ShanghaiTech Liu et al. (2018), and MSAD Zhu et al. (2024)—covering diverse real-world surveillance scenarios. Following standardized preprocessing and independent transformer encoding for each modality (see Appendix I), we perform evaluations using AUC and AP metrics. Ablation studies presented in Appendix J analyze the sensitivity of performance to key hyperparameters ($\nu$, $N$, $\lambda_{\text{ref}}$, and $\epsilon$) and investigate the effects of loss configuration choices (Eq. 7). Robustness to outlier injection and the behavior of uncertainty weights under perturbations are evaluated, with additional uncertainty-aware metrics, including KL divergence and Brier score, reported in Appendix K. Our method consistently demonstrates strong performance by effectively leveraging complementary spatial and temporal cues from image and motion-cue representations in a weakly supervised setting.

### 4.1 REAL-WORLD ANOMALY DETECTION IN SURVEILLANCE VIDEO

| Method | UCF-Crime AUC (%) | UCF-Crime Ano-AUC (%) | XD-Violence AP (%) | Shanghai-Tech AUC (%) | Method | MSAD AUC (%) |
|---|---|---|---|---|---|---|
| Sultani et al. Sultani et al. (2018) | 84.14 | 63.29 | 75.18 | 91.72 | MIST (I3D) Feng et al. (2021a) | 86.65 |
| Wu et al. Wu et al. (2020) | 84.57 | 62.21 | 80.00 | 95.24 | MIST (SwinT) Feng et al. (2021a) | 85.67 |
| AVVD Wu et al. (2022) | 82.45 | 60.27 | - | - | UR-DMU Zhou et al. (2023) | 85.02 |
| RTFM Tian et al. (2021) | 85.66 | 63.86 | 78.27 | 97.21 | UR-DMU (SwinT) Zhou et al. (2023) | 72.36 |
| UR-DMU Zhou et al. (2023) | 86.97 | 68.62 | 81.66 | 97.57 | MGFN (I3D) Chen et al. (2023b) | 84.96 |
| UMIL Lv et al. (2023) | 86.75 | 68.68 | - | 96.78 | MGFN (SwinT) Chen et al. (2023b) | 78.94 |
| VadCLIP Wu et al. (2024c) | 88.02 | 70.23 | 84.51 | 97.49 | TEVAD (I3D) Chen et al. (2023a) | 86.82 |
| STPrompt Wu et al. (2024b) | 88.08 | - | - | 97.81 | TEVAD (SwinT) Chen et al. (2023a) | 83.6 |
| OVVAD Wu et al. (2024a) | 86.40 | - | 66.53 | 96.98 | EGO Ding et al. (2025) | 87.36 |
| Fusion (Cross Attention) | 86.57 | 69.03 | 84.62 | 97.36 | Fusion (Cross Attention) | 92.07 |
| Fusion (Gating) | 87.86 | 69.74 | 84.62 | 97.71 | Fusion (Gating) | 92.75 |
| IEF-VAD (Gaussian) | 88.11 ± 0.28 | 70.48 ± 0.66 | 87.18 ± 0.55 | 97.91 ± 0.08 | IEF-VAD (Gaussian) | 92.27 ± 0.34 |
| IEF-VAD (Student-T) | **88.67** ± 0.45 | **71.50** ± 1.02 | **87.63** ± 0.54 | **97.98** ± 0.07 | IEF-VAD (Student-T) | **92.90** ± 0.27 |

Table 1: Comparison of various methods on multiple anomaly detection benchmarks, including UCF-Crime Sultani et al. (2018), XD-Violence Wu et al. (2020), ShanghaiTech Liu et al. (2018), and MSAD Zhu et al. (2024). All metrics are reported as the mean ($\pm 1$ standard deviation) of 10 runs.

Table 1 shows that *IEF-VAD* outperforms all prior weakly supervised detectors on every benchmark. On UCF-Crime Sultani et al. (2018), the Gaussian variant already matches the best published AUC

(88.11%), while the Student's t extension lifts AUC to 88.67% and raises the anomaly-focused AUC (Ano-AUC) to 71.50%—a $\approx 1.3$ pp absolute gain over the previous record (70.23% of VadCLIP Wu et al. (2024c)). Similar trends appear on XD-Violence Wu et al. (2020) (87.63 AP) and Shang-haiTech Liu et al. (2018) (97.98 AUC), where both variants surpass the strongest competitors. On the more recent MSAD Zhu et al. (2024) benchmark, our Student's-$t$ model achieves 92.90 AUC, exceeding the best I3D-based baseline (86.82 AUC) by over 6 pp. Fusion baselines such as cross-attention and gating exhibit modality dominance, with performance converging toward that of a single modality, whereas IEF-VAD consistently achieves higher scores. Standard-deviation margins indicate that gains are statistically consistent across 10 independent runs.

These results confirm two key insights of our framework. First, value-level uncertainty weighting enables practical exploitation of the heterogeneous synthetic motion cue, turning its complementary signals into measurable performance gains whenever the RGB stream is degraded. Second, modelling heavy-tailed noise via a Student's-$t$ likelihood yields further, systematic improvements, especially on long-tailed datasets (UCF-Crime Sultani et al. (2018), XD-Violence Wu et al. (2020)) where RGB frames frequently contain motion blur or illumination changes.

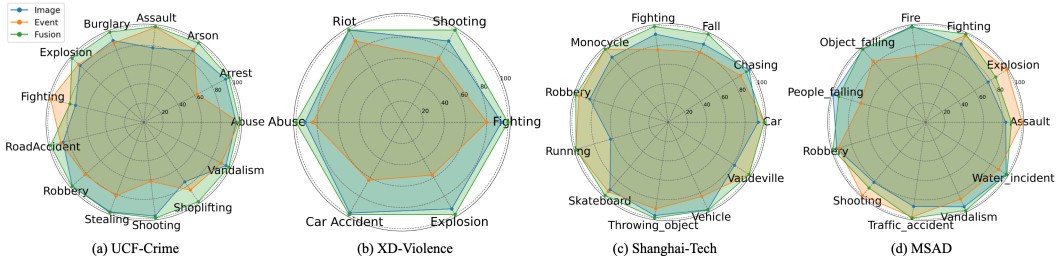

(a) UCF-Crime  (b) XD-Violence  (c) Shanghai-Tech  (d) MSAD

Figure 2: Radar charts showing per-class anomaly detection performance (AUC and AP) for image-only (blue), motion-cue-only (orange), and fused (green) approaches. Each radial axis represents an anomaly category, and values are normalized per class by the maximum score. The fused approach consistently covers a larger area, highlighting improved detection across anomaly types.

These trends are further illustrated in Figure 2, which provides a class-wise comparison of image-only, motion-cue-only, and fused approaches. In most anomaly classes, the image-based modality achieves higher performance than the motion-cue-based modality, reflecting the substantial differences in the underlying information each modality encodes. However, in anomaly categories that align more closely with the characteristics of $z_e$ (e.g., Fighting, Assault in UCF-Crime Sultani et al. (2018)), the motion-cue-based modality outperforms the image-based one, confirming our initial intuition (Figure 2). By fusing $z_x$ and $z_e$, our proposed method consistently achieves higher detection accuracy than either single modality alone. In particular, this fusion harnesses the strengths of image while absorbing the advantages of motion-cue for those classes where it excels, leading to further performance gains in categories already well-handled by image. Consequently, as illustrated in Figure 2, the fused approach covers a broader area in the radial plots, surpassing the capacity of single-modality baselines in most anomaly classes (see Appendix I for detailed numbers).

## 4.2 FUSION: UNCERTAINTY BEHAVIOR UNDER VALUE-LEVEL MASKING

To investigate how the uncertainty weights $w_m$ respond to modality-specific degradation, we perturb a single modality by masking a random subset of its latent features $z_m \in \mathbb{R}^{B \times T \times D}$. A masked sequence is defined as

$$z_{m,i}^{\text{masked}} = \begin{cases} 0, & i \in \mathcal{I}_\rho \\ z_{m,i}, & \text{otherwise} \end{cases}, \qquad |\mathcal{I}_\rho| = \rho D.$$

where $\rho \in (0,1)$ denotes the masking ratio and $i$ indexes feature dimensions. We then measure the change in the image-side uncertainty weight, $\Delta w_x = w_x^{\text{masked}} - w_x^{\text{clean}}$, with the normalisation $w_x + w_e = 1$ enforced. We prefer masking over additive noise because it produces a deterministic, localised degradation that preserves the Student-$t$ noise assumption and avoids the non-linear propagation artifacts that Gaussian perturbations introduce in attention layers, yielding a cleaner causal probe of the learned uncertainty mechanism.

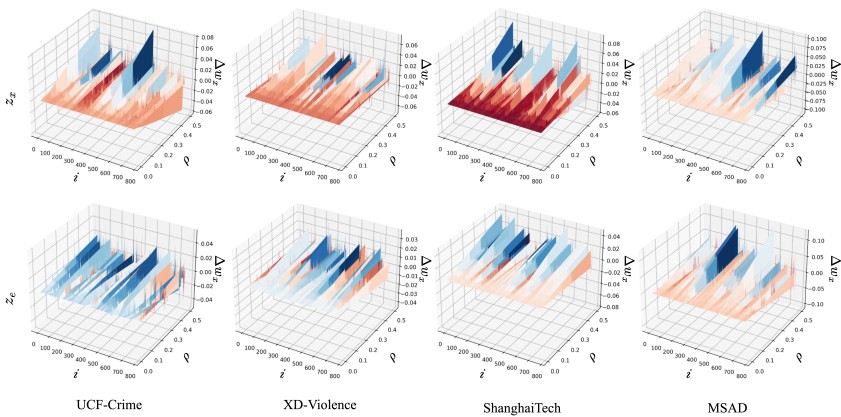

UCF-Crime  XD-Violence  ShanghaiTech  MSAD

Figure 3: Change in $\Delta w_x$ under modality-specific masking perturbations. The horizontal axis $i$ is the latent dimension index, and the vertical axis $\rho$ the proportion of masked values in $z_x$ or $z_e$. Each surface shows $\Delta w_x(i,\rho) = w_x^{\mathrm{masked}}(i) - w_x^{\mathrm{clean}}(i)$ for a given masking ratio. The top row applies masking to $z_x$, the bottom to $z_e$, both measuring the change in $w_x$. Positive values (blue) indicate increased image confidence under corruption, while negative values (red) indicate reduction. Non-uniform patterns across $i$ highlight dimension-specific responses to degradation.

Figure 3 plots $\Delta w_x$ for three masking ratios ($\rho \in 0.05, 0.10, 0.20$) on four datasets. The top row masks $z_x$, consistently decreasing $w_x$—indicating higher image uncertainty—whereas the bottom row masks $z_e$, symmetrically increasing $w_x$. The magnitude of $|\Delta w_x|$ grows with $\rho$, showing that the fusion network scales its confidence shift with corruption severity.

Value-level curves expose pronounced heterogeneity: some dimensions (e.g., indices 12–27) react strongly, whereas others remain flat. We attribute this to distributed encoding: dimensions dominated by stable appearance cues are robust, while those capturing transient, modality-specific dynamics are fragile. The non-uniform yet directionally consistent responses provide empirical evidence that the model estimates uncertainty on a fine-grained basis rather than collapsing it into a scalar. Supplementary statistics, including KL divergence and Brier score, are reported in Appendix K.

## 5 LIMITATIONS AND FUTURE WORK.

While our study advances uncertainty-guided fusion, several limitations remain. First, modeling each modality's noise with a diagonal covariance ignores cross-feature correlations; lightweight structured or low-rank estimators could better capture value-level dependencies. Second, fixed regularization weights $\lambda_{1,2}$ and Student-t degrees-of-freedom $\nu$ reduce adaptability; learning them jointly or via meta-learning would allow dynamic adjustment to modality quality. Third, replacing simple averaging with uncertainty-conditioned attention, gating, or message passing may yield a more expressive fusion mechanism that exploits inter-modal disagreement for stronger anomaly detection and broader multimodal perception.

## 6 CONCLUSION

IEF-VAD demonstrates that motion-centric, event-like representations—synthetically extracted from ordinary videos—can be fused with RGB frames to push video-anomaly detection beyond the limits of frame-based models, offering high practicality since they require no dedicated event cameras. The approach establishes new state-of-the-art scores on UCF-Crime, XD-Violence, ShanghaiTech, and MSAD, while a value-level masking study shows its weights shift adaptively across latent dimensions, revealing a reliable reliance on modality-specific cues. We expect these findings to catalyse broader use of synthetic motion-cue data and demonstrate its potential for principled, effective multimodal fusion in video understanding.

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

## A  SYNTHETIC MOTION CUE

We provide detailed explanations and analyses of the synthetic motion cue, which functions as a motion-centric proxy for event data within our framework.

**Generation.** Synthetic events are generated from consecutive RGB frames. Each frame $I_t$ is converted to grayscale, and inter-frame differences are computed as $\Delta_t(x, y) = |I_{t+1}(x, y) - I_t(x, y)|$. Pixels are activated when their difference exceeds a threshold, $E_t(x, y) = \mathbb{1}[|\Delta_t(x, y)| > \tau]$, and subsequently aggregated over a temporal window $T$: $\hat{E}(x, y) = \frac{1}{T} \sum_{t=1}^{T} E_t(x, y)$. This yields sparse binary maps that highlight localized motion while suppressing static background. Compared to optical flow, this procedure is lightweight, unsupervised, and requires no dense estimation or training.

**Motivation.** The goal of synthetic events is not to replicate the full functionality of neuromorphic Dynamic Vision Sensors (DVS), which provide polarity and microsecond-level temporal resolution. Instead, they extract motion-centric cues that are absent in static RGB appearance. These representations emphasize transient, localized dynamics that are crucial for anomaly detection while remaining scalable and deployable without specialized hardware. Thus, synthetic events complement RGB features by isolating motion saliency that often becomes diluted in appearance-only pipelines.

**Empirical Benefits.** On UCF-Crime, synthetic events alone substantially outperform RGB inputs on motion-sensitive classes such as *Assault* (56.44 → 72.03), *Fighting* (58.14 → 79.27), and *Shoplifting* (64.27 → 73.29). When fused with RGB under our uncertainty-aware formulation, performance is further improved (e.g., Shoplifting 73.29 → 85.72), demonstrating that synthetic events provide complementary cues rather than redundant information.

| Class | RGB | Synthetic Event |
|---|---|---|
| Assault | 56.44 | **72.03** |
| Fighting | 58.14 | **79.27** |
| Shoplifting | 64.27 | **73.29** |

Table 2: Synthetic events outperform RGB on motion-centric anomaly classes.

**Comparison with Real Events.** To validate their practicality, we compared synthetic events with real event streams from the UCF-Crime DVS benchmark. Fusion with real events improves Ano-AUC from 66.56 to 70.10, while our synthetic events achieve 72.22. This indicates that synthetic proxies, though simplified, serve as effective substitutes when real event sensors are unavailable.

| Method | AUC | Ano-AUC |
|---|---|---|
| Image only | 86.77 | 66.56 |
| Image + Real Event | 87.33 | 70.10 |
| Image + Synthetic Event (Ours) | **89.09** | **72.22** |

Table 3: Comparison between RGB, real events, and synthetic events on UCF-Crime.

**Robustness to Configuration.** We evaluated performance under varying temporal aggregation windows ($T \in \{4, 8, 16, 32, 64\}$) and motion thresholds. Across all settings, performance consistently remained above 88% AUC and 70% Ano-AUC, indicating that the method is not sensitive to configuration choices.

| TemporalAgg | 4 | 4 | 8 | 8 | 16 | 16 | 32 | 32 | 64 | 64 |
|---|---|---|---|---|---|---|---|---|---|---|
| Threshold | 10 | 25 | 10 | 25 | 10 | 25 | 10 | 25 | 10 | 25 |
| AUC | 89.00 | 88.91 | 89.26 | 89.44 | 89.09 | 89.55 | 88.83 | 89.58 | 88.40 | 89.33 |
| Ano-AUC | 72.13 | 72.18 | 72.63 | 73.24 | 72.22 | 73.38 | 71.50 | 73.40 | 70.36 | 72.70 |

Table 4: Performance across temporal aggregation and threshold variations.

**Robustness to Noise.** We further assessed resilience by injecting Poisson-based Background Activity Noise (BAN) into synthetic events. Performance degraded only marginally ($89.09 \rightarrow 87.89$ for $\lambda = 0.01$–$0.05$), showing stability under realistic sensor noise levels.

| BAN $\lambda$ | 0 | 0.01 | 0.02 | 0.03 | 0.04 | 0.05 | 0.1 | 0.2 | 0.3 | 0.4 | 0.5 |
|---|---|---|---|---|---|---|---|---|---|---|---|
| AUC | 89.09 | 88.43 | 88.09 | 87.99 | 87.87 | 87.89 | 87.62 | 87.11 | 86.71 | 86.71 | 86.83 |

Table 5: AUC degradation under simulated event noise (BAN).

**Attention and Interpretability.** Attention maps confirm the complementary role of synthetic events. RGB encoders typically cover 45–70% of the frame, shrinking only slightly during anomalies, which can dilute motion cues with background information. By contrast, event encoders consistently focus on compact 3–15% patches. On anomaly frames, event attention grows modestly (e.g., Assault $0.10 \rightarrow 0.14$), precisely localizing motion signals that RGB overlooks.

| Modality | Frame | Abuse | Arrest | Arson | Assault | Burglary | Explosion | Fighting | RoadAcc. | Robbery | Stealing | Shooting | Shoplifting | Vandalism |
|---|---|---|---|---|---|---|---|---|---|---|---|---|---|---|
| Image | Anomaly | 0.44 | 0.61 | 0.52 | 0.66 | 0.53 | 0.39 | 0.50 | 0.48 | 0.24 | 0.56 | 0.59 | 0.50 | 0.44 |
| Image | Normal | 0.56 | 0.67 | 0.46 | 0.71 | 0.51 | 0.47 | 0.51 | 0.52 | 0.34 | 0.51 | 0.60 | 0.52 | 0.45 |
| Event | Anomaly | 0.04 | 0.14 | 0.07 | 0.14 | 0.15 | 0.09 | 0.12 | 0.06 | 0.24 | 0.14 | 0.08 | 0.10 | 0.11 |
| Event | Normal | 0.03 | 0.05 | 0.04 | 0.10 | 0.10 | 0.06 | 0.09 | 0.09 | 0.09 | 0.07 | 0.05 | 0.09 | 0.05 |

Table 6: Class-wise attention area by modality and frame type. Event encoders attend to compact motion patches.

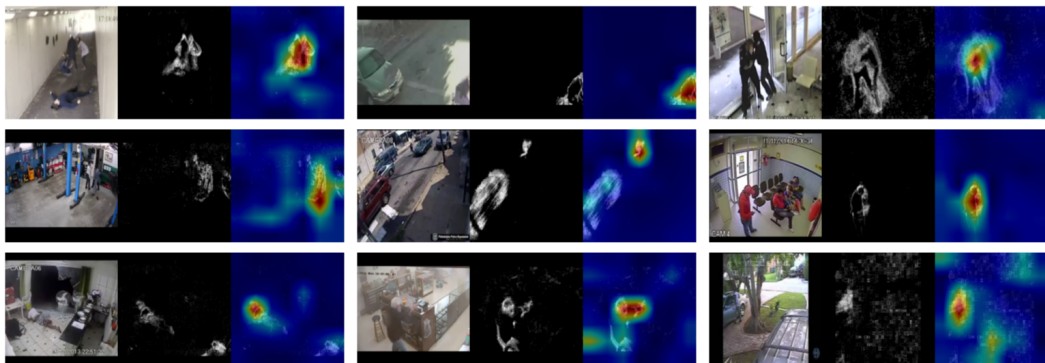

Figure 4: Qualitative comparison of RGB frames (left), synthetic event maps (middle), and attention heatmaps (right). The event encoder consistently focuses on compact motion-centric patches, while the RGB encoder distributes attention more broadly. This complementary behavior highlights why combining the two modalities improves anomaly detection.

Synthetic motion cues, though derived from RGB, act as effective event proxies. They (i) outperform RGB on motion-centric anomalies, (ii) complement appearance features, (iii) generalize to real events, (iv) remain robust to configuration changes and injected noise, and (v) provide interpretable attention footprints. These results confirm their practicality as substitutes for real event cameras in video anomaly detection.

# B   LIMITATIONS OF DIRECT STUDENT'S $t$ FUSION

## B.1   CLOSED-FORM FUSION INTRACTABILITY

The absence of a closed-form solution is the primary limitation of directly applying Student-$t$ distributions to precision-weighted fusion. Kalman-style fusion or Bayesian inverse-variance averaging relies on an explicit formulation such as:

$$\mu_f = \frac{w_1 \mu_1 + w_2 \mu_2}{w_1 + w_2}, \quad \text{where } w_i = \frac{1}{\sigma_i^2}.$$

This expression assumes that the product of two likelihoods remains within the same distribution family, allowing for analytic computation of fused mean and variance. While this holds for Gaussian distributions due to their closure under multiplication, it fails for the Student-$t$ family.

**Proof.** Let two independent Student-$t$ densities be

$$p_i(x) = \frac{1}{Z_i} \left( 1 + \frac{(x-\mu_i)^2}{\nu_i \sigma_i^2} \right)^{-\frac{\nu_i+1}{2}}, \quad i \in \{1, 2\}.$$

Their product is

$$p(x) \propto p_1(x)p_2(x) = \left( 1 + \frac{(x-\mu_1)^2}{\nu_1 \sigma_1^2} \right)^{-\frac{\nu_1+1}{2}} \left( 1 + \frac{(x-\mu_2)^2}{\nu_2 \sigma_2^2} \right)^{-\frac{\nu_2+1}{2}}.$$

This cannot be expressed as a single Student-$t$ law

$$p(x) \propto \left( 1 + \frac{(x-\mu)^2}{\tilde{\nu}\tilde{\sigma}^2} \right)^{-\frac{\tilde{\nu}+1}{2}}$$

because

- two distinct quadratic terms remain unless $\mu_1 = \mu_2$, $\nu_1 = \nu_2$, and $\sigma_1 = \sigma_2$;
- even then, the Student-$t$ family is not closed under multiplication.

Hence the fused posterior lacks closed-form moments, blocking Kalman updates. We address this by approximating each $t_{\nu_i}$ with a Gaussian using

$$\tilde{\sigma}_i^2 = \frac{\nu_i}{\nu_i + 1}\sigma_i^2,$$

restoring tractable inverse-variance fusion.

## B.2  UNSTABLE PRECISION

Inverse-variance fusion presumes a finite second moment, yet for a Student-$t_\nu$

$$\text{Var}(t_\nu) = \begin{cases} \frac{\nu}{\nu-2}\,\sigma^2, & \nu > 2, \\ \infty, & 1 < \nu \leq 2, \\ \text{undefined}, & \nu \leq 1. \end{cases}$$

Thus:

- For $\nu \leq 2$, the variance (and hence precision $w = 1/\sigma^2$) is non-finite, so inverse-variance weights cannot be defined.
- For $\nu > 2$ but near 2, the factor $\nu/(\nu - 2)$ explodes, making $w$ extremely sensitive to small changes in $\sigma^2$ and causing exploding or vanishing gradients.

These issues magnify in high-dimensional settings where per-feature noise accumulates, undermining stable training.

## B.3  GAUSSIAN SURROGATE VIA LAPLACE EXPANSION

To restore closed-form fusion we apply a Laplace (second-order) expansion at each mode:

$$t_\nu(\mu, \sigma^2) \approx \mathcal{N}(\mu, \tilde{\sigma}^2), \qquad \tilde{\sigma}^2 = \frac{\nu}{\nu+1}\sigma^2.$$

The Gaussian surrogate preserves local heavy-tail effects yet admits Kalman-style inverse-variance updates and a closed-form KL against a Gaussian prior. A derivation sketch is provided in the main text, while full details are presented here for completeness.

## C   BOUNDED INFLUENCE OF *Student's t* NOISE

**Proposition C.1** (Robustness of *Student's* t to Outliers)**.** *Let $\delta$ be a noise or residual term drawn from a univariate Student's t distribution with $\nu > 0$ degrees of freedom, location $0$, and scale $\sigma$:*

$$\delta \sim t_\nu(0, \sigma^2).$$

*The probability density function of $\delta$ is proportional to*

$$p(\delta) \;\propto\; \left(1 + \frac{\delta^2}{\nu\,\sigma^2}\right)^{-\frac{\nu+1}{2}}.$$

*Hence, the negative log-likelihood (omitting constant terms that do not depend on $\delta$) is*

$$-\log p(\delta) \;=\; \frac{\nu + 1}{2}\,\ln\!\left(1 + \frac{\delta^2}{\nu\,\sigma^2}\right) \;+\; \text{(constant)}.$$

*Differentiating with respect to $\delta$ yields the score function*

$$\frac{d}{d\delta}\big[-\log p(\delta)\big] \;=\; \frac{\nu+1}{2}\cdot\frac{1}{1 + \frac{\delta^2}{\nu\,\sigma^2}}\cdot\frac{2\,\delta}{\nu\,\sigma^2} \;=\; \frac{(\nu+1)\,\delta}{\nu\,\sigma^2 + \delta^2}.$$

*As $|\delta| \to \infty$, the denominator $\nu\,\sigma^2 + \delta^2$ is dominated by $\delta^2$, so*

$$\frac{(\nu+1)\,\delta}{\nu\,\sigma^2 + \delta^2} \;\approx\; \frac{(\nu+1)\,\delta}{\delta^2} \;=\; \frac{\nu+1}{\delta} \;\longrightarrow\; 0.$$

*Thus, the derivative (i.e., the slope or 'pull' of the residual on the log-likelihood) remains bounded and actually tends to zero for large outliers.*

*In contrast, consider a Gaussian noise model, $\delta \sim \mathcal{N}(0, \sigma^2)$. The corresponding negative log-likelihood is*

$$-\log p(\delta) \;=\; \frac{\delta^2}{2\,\sigma^2} \;+\; \text{(constant)},$$

*whose derivative is*

$$\frac{d}{d\delta}\big[-\log p(\delta)\big] \;=\; \frac{\delta}{\sigma^2},$$

which grows unboundedly as $|\delta| \to \infty$. Therefore, large outliers in a Gaussian model have much stronger influence on parameter estimation, making it less robust to extreme residuals.

Hence, the bounded slope in the Student-$t$ model demonstrates greater robustness against outliers: as $|\delta|$ becomes large, its influence on parameter updates (through gradient-based or maximum likelihood methods) remains finite. This property is central to why Student-$t$-based methods are often preferred in situations where occasional extreme values are expected.

## D   DIAGONAL COVARIANCE ASSUMPTION

We model uncertainty with a diagonal covariance structure, where each feature dimension is associated with an independent variance parameter. This reduces the computational burden compared to a full covariance matrix $\Sigma \in \mathbb{R}^{D \times D}$.

For a full covariance, computing the log-determinant $\log|\Sigma|$ and the quadratic form $(z - \mu)^\top \Sigma^{-1}(z - \mu)$ requires $\mathcal{O}(D^3)$ operations per sample. Under the diagonal assumption, these reduce to

$$\log|\Sigma| = \sum_{i=1}^{D} \log \sigma_i^2, \qquad (z - \mu)^\top \Sigma^{-1}(z - \mu) = \sum_{i=1}^{D} \frac{(z_i - \mu_i)^2}{\sigma_i^2},$$

both of which scale as $\mathcal{O}(D)$.

For each modality $m \in \{x, e\}$ and feature index $i$, the likelihood is modeled as

$$p(z_m[i] \mid \mu_m[i], \sigma_m^2[i], \nu) = t_\nu(\mu_m[i], \sigma_m^2[i]).$$

Applying a Laplace approximation yields an effective variance

$$\tilde{\sigma}_m^2[i] = \frac{\nu}{\nu+1}\sigma_m^2[i],$$

which enables computation of precision weights

$$w_m[i] = \frac{1}{\tilde{\sigma}_m^2[i] + \epsilon}, \qquad \mu_f[i] = \frac{w_x[i]\mu_x[i] + w_e[i]\mu_e[i]}{w_x[i] + w_e[i]}.$$

The diagonal covariance assumption makes per-dimension uncertainty estimation tractable and allows element-wise fusion without the prohibitive cost of full-covariance modeling.

## E   LAPLACE APPROXIMATION FOR THE *Student's t* NOISE MODEL

We derive an effective variance for the *Student's t* noise model via the *Laplace approximation*. Our goal is to approximate the heavy-tailed log-density by a quadratic (Gaussian) form in the vicinity of its mode, thereby obtaining an effective variance that scales the underlying Gaussian variance by a factor of $\nu/(\nu+1)$.

### STUDENT'S T DISTRIBUTION

The probability density function of the *Student's t* distribution with degrees of freedom $\nu$, location parameter $\mu$, and scale parameter $s > 0$ is given by

$$f(x; \nu, \mu, s) = \frac{\Gamma\left(\frac{\nu+1}{2}\right)}{s\sqrt{\nu\pi}\,\Gamma\left(\frac{\nu}{2}\right)}\left(1 + \frac{1}{\nu}\left(\frac{x-\mu}{s}\right)^2\right)^{-\frac{\nu+1}{2}},$$

where:

- $\Gamma(\cdot)$ denotes the Gamma function.
- $\nu > 0$ is the degrees of freedom, controlling the heaviness of the tails.
- $\mu \in \mathbb{R}$ is the location parameter (here, assumed to be zero in our derivation).
- $s > 0$ is the scale parameter; setting $\sigma^2 = s^2$ allows us to interpret $\sigma^2$ as the variance of the underlying Gaussian scale.

In our derivation we assume $\mu = 0$ for simplicity. Ignoring the normalization constant, the unnormalized density can then be written as

$$p(\delta) \propto \left(1 + \frac{\delta^2}{\nu\,\sigma^2}\right)^{-\frac{\nu+1}{2}},$$

where $\delta \in \mathbb{R}$ represents the noise term.

### DERIVATION VIA LAPLACE APPROXIMATION

We now present a detailed derivation of the effective variance via the *Laplace approximation*.

**Proposition E.1** (Effective Variance under the Laplace Approximation). *Let $\delta \in \mathbb{R}$ be a noise term with distribution (ignoring the normalization constant)*

$$p(\delta) \propto \left(1 + \frac{\delta^2}{\nu\,\sigma^2}\right)^{-\frac{\nu+1}{2}},$$

*where $\sigma^2$ is the scale (variance) parameter of the underlying Gaussian and $\nu > 0$ is the degrees of freedom. Then, by performing a second-order Taylor expansion of the log-density about its mode at $\delta = 0$, the local approximation is equivalent to that of a Gaussian distribution with effective variance*

$$\tilde{\sigma}^2 = \frac{\nu}{\nu+1}\sigma^2.$$

We start with the unnormalized density and take the natural logarithm to obtain the log-density as

$$p(\delta) \propto \left(1 + \frac{\delta^2}{\nu\,\sigma^2}\right)^{-\frac{\nu+1}{2}}, \quad \log p(\delta) = -\frac{\nu+1}{2} \log\left(1 + \frac{\delta^2}{\nu\,\sigma^2}\right) + C,$$

where $C$ is a constant independent of $\delta$.

Since the mode of $p(\delta)$ occurs at $\delta = 0$, we perform a Taylor expansion of $\log p(\delta)$ around $\delta = 0$. For small $x$, we have the approximation

$$x = \frac{\delta^2}{\nu\,\sigma^2}, \quad \log(1+x) \approx x \quad \text{(first-order Taylor expansion)}.$$

Thus, for small $\delta$,

$$\log\left(1 + \frac{\delta^2}{\nu\,\sigma^2}\right) \approx \frac{\delta^2}{\nu\,\sigma^2}.$$

Substituting this into the log-density expression yields

$$\log p(\delta) \approx -\frac{\nu+1}{2}\,\frac{\delta^2}{\nu\,\sigma^2} + C = -\frac{\nu+1}{2\nu\,\sigma^2}\,\delta^2 + C.$$

Now, consider the log-density of a Gaussian distribution with mean zero and variance $\tilde{\sigma}^2$:

$$\log p_G(\delta) = -\frac{1}{2\tilde{\sigma}^2}\,\delta^2 + C',$$

where $C'$ is a constant independent of $\delta$. To match the local quadratic approximation of $\log p(\delta)$, we equate the coefficients of $\delta^2$:

$$\frac{1}{2\tilde{\sigma}^2} = \frac{\nu+1}{2\nu\,\sigma^2}.$$

Multiplying both sides by 2 gives

$$\frac{1}{\tilde{\sigma}^2} = \frac{\nu+1}{\nu\,\sigma^2}.$$

Taking reciprocals, we obtain

$$\tilde{\sigma}^2 = \frac{\nu}{\nu+1}\,\sigma^2.$$

# F  THEORETICAL FOUNDATIONS OF INVERSE VARIANCE WEIGHTING IN BAYESIAN INFERENCE

In this appendix, we provide a detailed derivation and theoretical justification of the inverse variance (or precision) weighting scheme used in our fusion model. This method is firmly rooted in Bayesian inference, where each measurement contributes to the estimation of the latent variable based on its reliability.

## F.1  MEASUREMENT FUSION UNDER GAUSSIAN NOISE

Assume that we wish to estimate a latent variable $z$ from two independent noisy measurements $z_x$ and $z_e$. Each measurement is modeled as:

$$z_m = \mu_m + \delta_m, \quad \delta_m \sim \mathcal{N}(0, \sigma_m^2), \quad m \in \{x, e\}.$$

Here, $\mu_m$ represents the central estimate predicted by modality $m$, and $\sigma_m^2$ is the associated uncertainty (variance). The likelihood of observing $z_m$ given $z$ is then

$$p(z_m \mid z) \propto \exp\left(-\frac{(z_m - z)^2}{2\sigma_m^2}\right).$$

Assuming a flat prior $p(z)$, the posterior is proportional to the product of the likelihoods:

$$p(z \mid z_x, z_e) \propto p(z_x \mid z)\, p(z_e \mid z).$$

Taking the logarithm, we obtain the joint log-likelihood:

$$\log p(z \mid z_x, z_e) = -\frac{(z_x - z)^2}{2\sigma_x^2} - \frac{(z_e - z)^2}{2\sigma_e^2} + C.$$

Differentiating with respect to $z$ to find the maximum a posteriori (MAP) estimate $\hat{z}$:

$$\frac{\partial}{\partial z} \left[ -\frac{(z_x - z)^2}{2\sigma_x^2} - \frac{(z_e - z)^2}{2\sigma_e^2} \right] = 0, \quad \Rightarrow \quad \frac{z_x - z}{\sigma_x^2} + \frac{z_e - z}{\sigma_e^2} = 0.$$

Rearranging terms gives:

$$z \left( \frac{1}{\sigma_x^2} + \frac{1}{\sigma_e^2} \right) = \frac{z_x}{\sigma_x^2} + \frac{z_e}{\sigma_e^2},$$

and hence the fused estimate is:

$$\hat{z} = \frac{\frac{z_x}{\sigma_x^2} + \frac{z_e}{\sigma_e^2}}{\frac{1}{\sigma_x^2} + \frac{1}{\sigma_e^2}}.$$

This derivation shows that the optimal fusion under Gaussian noise is achieved by weighting each measurement with its inverse variance $w_m = \frac{1}{\sigma_m^2}$.

### F.2 BAYESIAN JUSTIFICATION VIA THE KALMAN FILTERING FRAMEWORK

The inverse variance weighting rule is further supported by the Bayesian update formulations seen in Kalman filtering. Consider a scenario where a prior estimate $\hat{z}^-$ with variance $\sigma^-$ is updated with a measurement $z_m$ having uncertainty $\sigma_m^2$. The Kalman update is given by:

$$\hat{z} = \hat{z}^- + K(z_m - \hat{z}^-),$$

where the Kalman gain $K$ is:

$$K = \frac{\sigma^-}{\sigma^- + \sigma_m^2}.$$

A measurement with lower uncertainty (higher precision) results in a larger Kalman gain, thus exerting a greater influence on the updated state. Extending this idea to the fusion of multiple modalities, the final fused estimate can be expressed as:

$$\mu_f = \frac{w_x \mu_x + w_e \mu_e}{w_x + w_e},$$

which is exactly the precision-weighted average obtained via the MAP estimation under the assumed likelihood models.

### F.3 FUSION FORMULA: DUAL THEORETICAL FOUNDATIONS

Our fusion formula is derived based on two theoretical foundations: Bayesian inference and the Kalman filter. First, assume that the two modalities provide independent estimates of the same latent variable $z$. For the Gaussian case, the estimates are given by

$$p(z \mid \mu_i, \sigma_i^2) = \mathcal{N}(z; \mu_i, \sigma_i^2) \quad \text{and} \quad p(z \mid \mu_e, \sigma_e^2) = \mathcal{N}(z; \mu_e, \sigma_e^2).$$

Because these estimates are independent, the joint likelihood (or the unnormalized posterior under a uniform prior) is proportional to the product of the two Gaussians:

$$p(z \mid \mu_i, \mu_e) \propto \exp\left( -\frac{||z - \mu_i||^2}{2\sigma_i^2} \right) \cdot \exp\left( -\frac{||z - \mu_e||^2}{2\sigma_e^2} \right).$$

By combining the exponents and completing the square, we find that the value of $z$ that maximizes the posterior—i.e., the fused mean—is given by

$$\mu_f = \frac{\mu_i/\sigma_i^2 + \mu_e/\sigma_e^2}{1/\sigma_i^2 + 1/\sigma_e^2}.$$

For the *Student's* $t$ model, we replace $\sigma_i^2$ and $\sigma_e^2$ with their effective counterparts $\tilde{\sigma}_i^2$ and $\tilde{\sigma}_e^2$, as derived via the Laplace approximation (see below), leading to the same formulation in terms of the inverse variances.

From the perspective of the Kalman filter, suppose that one modality provides a prediction $\mu_i$ with variance $\sigma_i^2$ (or $\tilde{\sigma}_i^2$ in the *Student's t* case) and another modality provides a prediction $\mu_e$ with variance $\sigma_e^2$ (or $\tilde{\sigma}_e^2$ in the *Student's t* case). The Kalman filter update for the state estimates is given by

$$\mu_f = \mu_i + K(\mu_e - \mu_i),$$

where the Kalman gain $K$ is defined as

$$K = \frac{\tilde{\sigma}_i^2}{\tilde{\sigma}_i^2 + \tilde{\sigma}_e^2},$$

which leads to an equivalent expression for the fused mean:

$$\mu_f = \frac{\tilde{\sigma}_e^2 \mu_i + \tilde{\sigma}_i^2 \mu_e}{\tilde{\sigma}_i^2 + \tilde{\sigma}_e^2}.$$

By defining the inverse variance weights as $w_i = 1/(\sigma_i^2 + \epsilon)$ (or $w_i = 1/(\tilde{\sigma}_i^2 + \epsilon)$ for the *Student's t*) and $w_e = 1/(\sigma_e^2 + \epsilon)$ (or $w_e = 1/(\tilde{\sigma}_e^2 + \epsilon)$ for the *Student's t*), our fusion formula becomes

$$\mu_f = \frac{w_i \mu_i + w_e \mu_e}{w_i + w_e}.$$

Thus, both the Bayesian derivation and the Kalman filter interpretation lead to the same uncertainty-weighted fusion formula, with the only difference being that for the *Student's t* noise model we use a corrected (effective) variance.

To summarize, our fusion formula is derived based on two theoretical foundations: Bayesian inference and the Kalman filter. Both derivations lead to the same uncertainty-weighted fusion expression:

$$\mu_f = \frac{w_i \mu_i + w_e \mu_e}{w_i + w_e},$$

with the inverse variance weights defined as $w_m = 1/(\sigma_m^2 + \epsilon)$ for Gaussian noise and $w_m = 1/(\tilde{\sigma}_m^2 + \epsilon)$ for the *Student's t* model. This dual theoretical basis justifies our approach, as it effectively leverages the inverse variances (or precisions) of the modality-specific estimates to account for their respective uncertainties, resulting in a robust and reliable fused representation for downstream tasks.

## G    ITERATIVE REFINEMENT OF FUSED STATE

After performing the sequential update to fuse the modalities over time using effective variances (i.e.,

$$\tilde{\sigma}^2 = \exp\left(\log \sigma^2 + \log\left(\frac{\nu}{\nu + 1}\right)\right)$$

) to account for the heavy-tailed Student-T noise, small residual errors or microscopic uncertainties may still persist in the fused state. To address this, we introduce an iterative refinement step that further denoises and adjusts the fused representation.

The intuition behind this approach is similar to iterative error correction or gradient descent-based optimization: rather than relying solely on the initial fusion, we continuously refine the estimate to better capture the true latent state. Specifically, starting from the initial fused state $x_{\text{fusion}}^0$ obtained after the sequential update, we iteratively predict and subtract a residual correction. In each refinement step $i$ (for $i = 0, 1, \ldots, N - 1$), a dedicated network $F(\cdot)$ takes the current state $x^i$ along with additional contextual information $c_i$ (which may include time-step context, current effective uncertainty estimates, and modality weights) and predicts a residual $\Delta x^i$:

$$\Delta x^i = F(x^i, c_i).$$

This residual represents the remaining error in the current fused estimate. The state is then updated by subtracting a fraction of this residual, controlled by an attenuation parameter $\lambda_i$:

$$x^{i+1} = x^i - \lambda_i \Delta x^i.$$

After $N$ refinement steps, the final fused representation $x^N$ is obtained, which is expected to be more robust and accurate.

**Theoretical Justification:** Even after uncertainty-weighted fusion and sequential updates—where effective variances derived via the Student-T model are used—the fused state may still contain imperfections due to noise, model approximation errors, or unmodeled dynamics. The refinement network is motivated by the following principles:

- **Residual Learning:** The initial fused state $x^0_{\text{fusion}}$ is an approximation of the true latent state $y$, such that

$$y = x^0_{\text{fusion}} + \delta,$$

where $\delta$ is the residual error. The refinement network is designed to learn this residual:

$$F^*(x^i, c_i) \approx \mathbb{E}[y - x^i \mid x^i, c_i],$$

so that the final estimate becomes

$$x^N = x^0_{\text{fusion}} - \sum_{i=0}^{N-1} \lambda_i F(x^i, c_i).$$

This formulation is analogous to residual learning in deep networks, where modeling the error is often easier than directly predicting the target.

- **Diffusion Model Inspiration:** Diffusion models iteratively denoise data by progressively removing noise from a corrupted input. Similarly, our iterative refinement can be viewed as a denoising process where each refinement step removes part of the residual error, thereby driving the fused state closer to the true latent representation.

- **Optimization Perspective:** The refinement step can be interpreted as performing an additional optimization in function space. The subtraction of $\lambda_i \Delta x^i$ is akin to a gradient descent update that reduces an implicit error loss. Over multiple iterations, this results in a more accurate estimate, provided that the refinement network is properly designed and trained.

**Empirical Benefits and Future Directions:** In practice, introducing a refinement network after the initial fusion offers several benefits:

- **Error Reduction:** By learning the residual $\delta$, the final output $x^N$ achieves lower prediction error than the initial fused state.

- **Robustness:** The iterative refinement is effective in mitigating the effects of heavy-tailed noise and unmodeled dynamics, leading to a more stable fused representation.

- **Enhanced Detail Recovery:** Fine-grained details that might be lost in the initial fusion can be recovered through successive refinement, improving both quantitative metrics and qualitative performance.

Future work may explore:

- **Iterative or Recurrent Refinement:** Extending the refinement process with additional iterative steps or a recurrent architecture that shares weights across iterations.

- **Uncertainty-Guided Refinement:** Incorporating explicit uncertainty measures to guide the refinement network to focus on regions with high residual error.

- **Enhanced Loss Functions:** Employing perceptual or adversarial losses in the refinement stage to better capture fine details and enhance the realism of the final output.

In summary, the iterative refinement network not only removes residual errors remaining after the initial uncertainty-weighted fusion but also draws strong inspiration from diffusion models' denoising principles. This two-stage approach—first, a coarse fusion and then a fine, iterative refinement—provides a theoretically grounded and empirically validated method to enhance the final fused representation for downstream tasks.

## H  DERIVATION OF KL DIVERGENCE

We present a detailed derivation of the KL divergence between two Gaussian distributions. Specifically, the KL divergence between a Gaussian distribution $\mathcal{N}(\mu_m, \tilde{\sigma}_m^2)$ and a standard normal distribution $\mathcal{N}(0, I)$ is derived in closed form as follows:

$$KL\big(\mathcal{N}(\mu_m, \tilde{\sigma}_m^2)\|\mathcal{N}(0, I)\big) = \frac{1}{2}(\tilde{\sigma}_m^2 + \mu_m^2 - 1 - \log\tilde{\sigma}_m^2).$$

The KL divergence between two probability distributions $p(x)$ and $q(x)$ is defined as:

$$KL(p\|q) = \int p(x)\log\frac{p(x)}{q(x)}\,dx.$$

Consider two Gaussian distributions:

$$p(x) = \mathcal{N}(x; \mu_m, \tilde{\sigma}_m^2), \quad q(x) = \mathcal{N}(x; 0, 1).$$

Expanding explicitly, we obtain:

$$KL(p\|q) = \int p(x)\left[\log\frac{1}{\sqrt{2\pi\tilde{\sigma}_m^2}}\exp\left(-\frac{(x-\mu_m)^2}{2\tilde{\sigma}_m^2}\right) - \log\frac{1}{\sqrt{2\pi}}\exp\left(-\frac{x^2}{2}\right)\right]dx.$$

$$KL(p\|q) = \frac{1}{2}\log\frac{1}{\tilde{\sigma}_m^2} + \int p(x)\left[-\frac{(x-\mu_m)^2}{2\tilde{\sigma}_m^2} + \frac{x^2}{2}\right]dx.$$

Evaluating the expectations under $p(x)$:

$$\mathbb{E}_p[x] = \mu_m, \quad \mathbb{E}_p[x^2] = \tilde{\sigma}_m^2 + \mu_m^2, \quad \mathbb{E}_p[(x-\mu_m)^2] = \tilde{\sigma}_m^2.$$

Substituting these expectations back into the expression gives:

$$KL(p\|q) = \frac{1}{2}\log\frac{1}{\tilde{\sigma}_m^2} - \frac{1}{2\tilde{\sigma}_m^2}\tilde{\sigma}_m^2 + \frac{1}{2}(\tilde{\sigma}_m^2 + \mu_m^2).$$

Simplifying further, we obtain the final closed-form expression:

$$KL\big(\mathcal{N}(\mu_m, \tilde{\sigma}_m^2)\|\mathcal{N}(0, I)\big) = \frac{1}{2}(\tilde{\sigma}_m^2 + \mu_m^2 - 1 - \log\tilde{\sigma}_m^2).$$

## I  EXPERIMENT DETAILS

**Experiments compute resources.** All experiments were conducted on a local workstation equipped with an AMD Ryzen Threadripper PRO 5955WX 16-Core Processor (32 threads) and a single NVIDIA RTX 6000 Ada Generation GPU (48GB VRAM). The system had 256GB of system RAM and 5GB VRAM and ran on Ubuntu 22.04. To improve training efficiency, we first precompute the video frame embeddings using the image and event encoders before training. This preprocessing step takes approximately 3 to 5 days. Once embeddings are extracted, training on the XD-Violence dataset takes around 3 hours, while training on the ShanghaiTech, UCF-Crime, and MSAD datasets completes within 1 hour.

### I.1  DATASET DETAILS

We evaluate our weakly supervised learning approach on four commonly used benchmark datasets for video anomaly detection: UCF-Crime Sultani et al. (2018), XD-Violence Wu et al. (2020), ShanghaiTech Liu et al. (2018), and MSAD Zhu et al. (2024). UCF-Crime consists of 1,900 real surveillance videos, totaling 128 hours and covering 13 anomaly classes. XD-Violence comprises 4,754 untrimmed videos (217 hours), featuring 6 distinct anomalous or violent actions. ShanghaiTech includes 330 training and 107 test videos (approximately 317,000 frames), recorded in 13 scenes and labeled with 11 anomaly classes. MSAD features 720 videos from 14 different scenarios, annotated with 11 anomalies. Due to the data imbalance and the rarity of violent incidents in XD-Violence, we report the Average Precision (AP, %) to assess precision–recall balance, while for the other datasets, we measure performance using the Area Under the ROC Curve (AUROC, %).

**Preprocessing** We employ a spatial augmentation strategy inspired by multi-crop evaluation. Specifically, each input video is first resized to a resolution of $224 \times 224$, followed by the generation of 10 spatial crops: five fixed regions (top-left, top-right, bottom-left, bottom-right, and center) and their horizontally flipped counterparts. From each video, image frames are extracted and processed by the CLIP Radford et al. (2021) (ViT-L/14) image encoder to obtain latent representations. For every 16 frames, we compute the average latent vector, denoted as $z_x$. In parallel, we generate *synthetic motion cues* by computing pixel-wise changes between consecutive frames within each 16-frame segment using a threshold of 10/255 and a clamp value of 10. The resulting binary motion maps—serving as *event-like proxies*—are stacked and fed into the event encoder Jeong et al. (2024) (aligned with the image encoder) to produce motion representations $z_e$. We refer to these as "synthetic" since they are derived from RGB frames rather than captured by a true event sensor.

## I.2 IMPLEMENTATION DETAILS

**Architecture Detail** Figure 1 illustrates the complete architecture used in our framework. Both the image and event encoders are implemented using the ViT-L/14 architecture from CLIP Radford et al. (2021), with an embedding dimension of 768. The function $f$, responsible for modality-specific feature encoding, is implemented as a multi-layer attention module with 8 attention heads and 2 transformer layers. The functions $g$ and $h$, used to predict the mean and variance parameters for fusion, are each implemented as a single linear layer. The refinement network consists of a simple feedforward structure with a Linear–ReLU–Linear sequence to iteratively refine the fused representation.

**Hyperparameters Detail** To reproduce the results reported in Table 1, we configure the model with the following hyperparameters: the degrees of freedom is set to $\nu=8$, the number of iterative refinement steps is $N=10$, and the uncertainty refinement weight is $\lambda_r=0.5$. The optimization uses the AdamW Loshchilov & Hutter (2017) optimizer with a learning rate of $2 \times 10^{-5}$ and a batch size of 64 for 10 training epochs. We apply a MultiStepLR scheduler with milestones at epochs 4 and 8, and a decay factor of 0.1. The numerical stability term is set to $\epsilon=10^{-8}$. Additionally, for the regularization loss $\mathcal{L}_{\text{reg}}$, we fix both $\lambda_1$ and $\lambda_2$ to 0.5 throughout all experiments.

## I.3 RESULTS DETAIL

| **UCF-Crime** Sultani et al. (2018) | | | | **XD-Violence** Wu et al. (2020) | | | | **ShanghaiTech** Liu et al. (2018) | | | | **MSAD** Zhu et al. (2024) | | | |
|---|---|---|---|---|---|---|---|---|---|---|---|---|---|---|---|
| Class | Image | Event | Fusion | Class | Image | Event | Fusion | Class | Image | Event | Fusion | Class | Image | Event | Fusion |
| Abuse | 68.02 | 70.09 | 70.74 | Fighting | 79.59 | 67.81 | 84.76 | Car | 70.07 | 74.76 | 74.83 | Assault | 54.78 | 66.03 | 58.73 |
| Arrest | 72.21 | 47.09 | 75.05 | Shooting | 54.59 | 42.94 | 61.99 | Chasing | 94.49 | 84.36 | 91.33 | Explosion | 50.86 | 66.25 | 57.20 |
| Arson | 65.49 | 66.75 | 72.68 | Riot | 97.62 | 86.07 | 97.67 | Fall | 72.96 | 65.30 | 82.17 | Fighting | 71.74 | 79.75 | 81.14 |
| Assault | 56.44 | 72.03 | 72.58 | Abuse | 59.42 | 54.49 | 64.96 | Fighting | 76.91 | 63.48 | 83.64 | Fire | 71.97 | 49.44 | 71.23 |
| Burglary | 68.02 | 65.88 | 74.99 | Car Accident | 50.83 | 32.53 | 51.70 | Monocycle | 67.23 | 75.32 | 75.46 | Object_falling | 90.52 | 75.92 | 90.92 |
| Explosion | 56.33 | 57.64 | 63.46 | Explosion | 64.32 | 39.22 | 68.54 | Robbery | 76.74 | 87.26 | 90.43 | People_falling | 60.64 | 42.50 | 56.93 |
| Fighting | 58.14 | 79.27 | 62.81 | | | | | Running | 37.95 | 60.95 | 60.78 | Robbery | 68.10 | 66.90 | 70.95 |
| RoadAccident | 57.41 | 59.11 | 66.32 | | | | | Skateboard | 76.04 | 78.29 | 82.38 | Shooting | 71.20 | 86.87 | 77.88 |
| Robbery | 76.03 | 62.39 | 76.29 | | | | | Throwing_object | 89.63 | 83.14 | 91.95 | Traffic_accident | 62.23 | 70.08 | 70.93 |
| Stealing | 74.91 | 61.51 | 75.43 | | | | | Vehicle | 79.39 | 67.38 | 79.87 | Vandalism | 83.40 | 75.82 | 87.05 |
| Shooting | 60.95 | 38.42 | 62.26 | | | | | Vaudeville | 44.04 | 53.66 | 53.61 | Water_incident | 97.95 | 88.93 | 98.75 |
| Shoplifting | 64.27 | 73.29 | 85.72 | | | | | | | | | | | | |
| Vandalism | 66.89 | 63.05 | 69.23 | | | | | | | | | | | | |
| AUC | 86.77 | 78.67 | 89.13 | AP | 84.22 | 55.96 | 86.54 | AUC | 97.58 | 93.69 | 98.24 | AUC | 91.52 | 82.10 | 92.16 |
| Ano AUC | 66.56 | 63.94 | 72.49 | | | | | | | | | | | | |

Table 7: Per-class performance on UCF-Crime Sultani et al. (2018), XD-Violence Wu et al. (2020), ShanghaiTech Liu et al. (2018), and MSAD Zhu et al. (2024) for Figure 2. Event is our event-like proxy (motion cue)

Figure2 summarizes the per-class performance across four benchmark datasets: UCF-Crime, XD-Violence, ShanghaiTech, and MSAD. For each anomaly class, we report detection performance using the image modality, motion-cue modality, and their fusion. In most cases, the fusion consistently outperforms both unimodal inputs, highlighting the complementarity between image and motion-cue information. Notably, on UCF-Crime, classes such as Shoplifting and Arson show substantial gains from fusion. For XD-Violence, categories like Riot and Fighting exhibit strong improvements with fusion, despite relatively weak motion-cue-only performance. ShanghaiTech also shows a consistent pattern of fusion superiority across diverse scene types. In the MSAD dataset, fusion leads to higher detection scores for complex dynamic events such as Water_incident and Vandalism. These

results emphasize the effectiveness of our uncertainty-guided multimodal fusion strategy in adapting to diverse scene contexts and anomaly types.

## J  ABLATION STUDY

### J.1  SENSITIVITY TO HYPERPARAMETER SETTINGS

We conduct a comprehensive ablation study to examine the effect of key hyperparameters in our uncertainty-guided fusion framework. Specifically, we analyze the impact of the degrees of freedom $\nu$ in the Student-T distribution, the number of refinement steps $N$, the Laplace approximation precision $\epsilon$, and the refinement weight $\lambda_r$, all metrics is reported 10-times average with 1-standard deviation.

| $\nu$ | **2** | **4** | **6** | **8** | **10** |
|---|---|---|---|---|---|
| AUC | 87.98±0.33 | 88.34±0.30 | 88.38±0.40 | 88.67±0.45 | 88.11±0.47 |
| Ano-AUC | 70.06±0.72 | 70.69±0.72 | 70.88±0.86 | 71.50±1.02 | 70.22±0.86 |
| **N** | **0** | **10** | **20** | **30** | **40** |
| AUC | 86.77±0.18 | 88.67±0.45 | 88.41±0.43 | 88.48±0.28 | 88.60±0.31 |
| Ano-AUC | 67.41±0.43 | 71.50±1.02 | 71.27±0.61 | 71.36±0.59 | 71.45±0.58 |
| $\epsilon$ | $10^{-4}$ | – | $10^{-6}$ | – | $10^{-8}$ |
| AUC | 88.46±0.42 | – | 88.50±0.38 | – | 88.67±0.45 |
| Ano-AUC | 71.06±1.14 | – | 71.45±0.81 | – | 71.50±1.02 |
| $\lambda_r$ | **0.1** | **0.3** | **0.5** | **0.7** | **0.9** |
| AUC | 87.92±0.38 | 88.13±0.24 | 88.67±0.45 | 88.21±0.34 | 87.91±0.34 |
| Ano-AUC | 69.61±0.98 | 70.13±0.58 | 71.50±1.02 | 70.32±0.90 | 69.80±0.86 |

Table 8: Ablation study results on UCF-Crime with varying hyperparameters.

**UCF-Crime** : Varying the degrees of freedom $\nu$ shows a gradual increase in both AUC and Ano-AUC scores, peaking at $\nu = 8$. For the number of refinement steps $N$, performance increases significantly when $N$ changes from 0 to 10, and remains relatively stable for $N \geq 10$. The Laplace approximation precision $\epsilon$ results in only marginal differences across all values tested. In the case of the refinement weight $\lambda_r$, the best results are obtained at $\lambda_r = 0.5$, while performance slightly degrades when the weight is set to more extreme values (0.1 or 0.9) Table 8.

| Dataset | Hyperparameter | 2 / 0 / 1e-4 / 0.1 | 4 / 10 / – / 0.3 | 6 / 20 / 1e-6 / 0.5 | 8 / 30 / – / 0.7 | 10 / 40 / 1e-8 / 0.9 |
|---|---|---|---|---|---|---|
| XD-Violence | $\nu$ (AP) | 87.31±0.77 | 87.39±0.65 | 87.03±0.52 | 87.63±0.54 | 86.64±0.51 |
| | N (AP) | 86.01±1.10 | 87.63±0.54 | 87.52±0.40 | 87.71±0.70 | 87.54±0.60 |
| | $\epsilon$ (AP) | 87.27±0.76 | – | 87.44±0.79 | – | 87.63±0.54 |
| | $\lambda_r$ (AP) | 86.89±0.80 | 86.74±0.48 | 87.63±0.54 | 87.19±0.67 | 87.23±1.13 |
| ShanghaiTech | $\nu$ (AUC) | 97.91±0.10 | 97.91±0.10 | 97.90±0.10 | 97.98±0.07 | 97.92±0.06 |
| | N (AUC) | 97.94±0.08 | 97.98±0.07 | 97.90±0.12 | 97.90±0.11 | 97.93±0.07 |
| | $\epsilon$ (AUC) | 97.91±0.12 | – | 97.90±0.08 | – | 97.98±0.07 |
| | $\lambda_r$ (AUC) | 97.86±0.07 | 97.90±0.11 | 97.98±0.07 | 97.90±0.11 | 97.92±0.13 |
| MSAD | $\nu$ (AUC) | 91.78±0.34 | 91.79±0.29 | 91.84±0.42 | 92.90±0.27 | 91.64±0.27 |
| | N (AUC) | 90.69±0.35 | 92.90±0.27 | 92.29±0.30 | 92.72±0.29 | 92.69±0.21 |
| | $\epsilon$ (AUC) | 91.77±0.27 | – | 91.81±0.33 | – | 92.90±0.27 |
| | $\lambda_r$ (AUC) | 90.73±0.74 | 91.41±0.53 | 92.90±0.27 | 92.12±0.29 | 92.28±0.44 |

Table 9: Ablation study results across datasets (XD-Violence, ShanghaiTech, MSAD) under varying hyperparameters.

**XD-Violence** : In the XD-Violence dataset, the degrees of freedom $\nu$ show stable performance across different settings, with AP values consistently around 87, peaking at 87.63 when $\nu = 8$. Increasing the number of refinement steps $N$ leads to noticeable improvements, with AP rising from

86.01 at $N = 2$ to 87.71 at $N = 30$. The Laplace precision parameter $\epsilon$ exhibits minimal influence on performance, with AP differences within 0.36 across tested values. Adjusting the refinement weight $\lambda_r$ shows moderate effects, where the best AP of 87.63 is achieved at $\lambda_r = 0.5$ Table 9.

**ShanghaiTech** : For the ShanghaiTech dataset, the performance remains stable across different settings of the degrees of freedom $\nu$, with AUC scores ranging narrowly between 97.90 and 97.98. Varying the number of refinement steps $N$ does not lead to significant changes, though a slight increase is observed at $N = 10$. The Laplace precision parameter $\epsilon$ shows minimal effect on AUC, with differences remaining within 0.08. Adjusting the refinement weight $\lambda_r$ yields the highest AUC of 97.98 at $\lambda_r = 0.5$, while other values produce slightly lower but comparable performance Table 9.

**MSAD** : In the MSAD dataset, increasing the degrees of freedom $\nu$ generally results in marginal improvements, peaking at $\nu = 8$ with an AUC of 92.90. The number of refinement steps $N$ has a more pronounced impact, with AUC improving steadily from 90.69 at $N = 0$ to 92.72 at $N = 30$, then maintaining a similar level at $N = 40$. The Laplace precision parameter $\epsilon$ again leads to only small variations, with values ranging from 91.77 to 92.90. For $\lambda_r$, AUC increases consistently as the parameter increases, with the highest value (92.28) observed at $\lambda_r = 0.9$ Table 9.

## J.2 LOSS CONFIGURATION

| Dataset | Metric | $\mathcal{L}_{\mathbf{cls}}$ | $\mathcal{L}_{\mathbf{cls}} + \mathcal{L}_{\mathbf{reg}}$ | $\mathcal{L}_{\mathbf{cls}} + \mathcal{L}_{\mathbf{KL}}$ | $\mathcal{L}_{\mathbf{cls}} + \mathcal{L}_{\mathbf{reg}} + \mathcal{L}_{\mathbf{KL}}$ |
|---|---|---|---|---|---|
| UCF-Crime | AUC | 88.04±0.28 | 87.78±0.46 | 88.24±0.19 | 88.67±0.45 |
| | Ano-AUC | 69.79±0.85 | 69.73±1.00 | 70.46±0.69 | 71.50±1.02 |
| XD-Violence | AP | 87.40±0.72 | 87.45±0.49 | 87.46±0.64 | 87.63±0.54 |
| ShanghaiTech | AUC | 97.97±0.07 | 97.91±0.09 | 97.85±0.14 | 97.98±0.07 |
| MSAD | AUC | 92.39±0.22 | 91.65±0.32 | 92.38±0.22 | 92.90±0.27 |

Table 10: Performance comparison under different loss configurations across datasets.

We assess the contribution of each loss component by evaluating four configurations: classification loss only ($\mathcal{L}_{\mathbf{cls}}$), classification plus regularization ($\mathcal{L}_{\mathbf{cls}} + \mathcal{L}_{\mathbf{reg}}$), classification plus KL divergence ($\mathcal{L}_{\mathbf{cls}} + \mathcal{L}_{\mathbf{KL}}$), and the full combination ($\mathcal{L}_{\mathbf{cls}} + \mathcal{L}_{\mathbf{reg}} + \mathcal{L}_{\mathbf{KL}}$) across UCF-Crime, XD-Violence, ShanghaiTech, and MSAD datasets (Table 10).

In UCF-Crime, adding the KL loss improves AUC from 88.04% to 88.24% and Ano-AUC from 69.79% to 70.46%, while the full combination further boosts AUC and Ano-AUC to 88.67% and 71.50%, respectively. In XD-Violence, ShanghaiTech, and MSAD, performance remains largely stable across settings, with slight improvements under the full loss configuration. In particular, MSAD AUC increases from 92.39% to 92.90% with the full loss. Overall, adding $\mathcal{L}_{\mathbf{KL}}$ consistently yields benefits, while the effect of $\mathcal{L}_{\mathbf{reg}}$ alone is minor. The full configuration achieves the best or comparable results across all benchmarks.

## K FUSION DETAILS

To evaluate whether the proposed framework performs as intended, we analyze the behavior of the uncertainty weights $w_m$ (Eq. 4) when one modality is partially corrupted. Specifically, we aim to test whether the model's dynamic uncertainty estimation mechanism can correctly respond to degraded sensor input. Rather than injecting additive noise—which may lead to complex and unpredictable interactions within attention-based encoders—we opt for a controlled masking strategy. In attention networks, even small perturbations can propagate nonlinearly across dimensions, making it difficult to interpret the resulting change in uncertainty due to entangled feature dependencies. Additionally, adversarial perturbations rely on gradients computed after the refinement stage, making it difficult to isolate the direct effect on modality-specific uncertainty. In contrast, masking fixed proportions of input features allows us to deterministically degrade the modality in a localized and interpretable manner, providing a clean testbed for evaluating the reliability and sensitivity of uncertainty estimation.

To assess the reliability of our uncertainty-weighted fusion mechanism, we conduct controlled modality-specific perturbation experiments across four datasets. We simulate degradation by randomly masking a proportion $\rho \in \{0.05, 0.10, 0.20, 0.30, 0.50\}$ of the latent feature dimensions in either the image modality ($z_x$) or the motion-cue modality ($z_e$). All experiments report standard performance metrics, including AUC or AP for detection quality, Brier score for probabilistic calibration, and KL divergence to quantify the shift between predictions made on clean inputs and those made under masking. Additionally, we track uncertainty weights $w_x$ and $w_e$ for both modalities, including breakdowns on abnormal and normal video segments, to understand how the model reallocates modality-level confidence under degradation.

| Noise Type | Masked Level | AUC (%) | Brier | KL | $\Delta w_e$ | $\Delta w_e^{\mathrm{ab}}$ | $\Delta w_e^{\mathrm{n}}$ | $\Delta w_x$ | $\Delta w_x^{\mathrm{ab}}$ | $\Delta w_x^{\mathrm{n}}$ |
|---|---|---|---|---|---|---|---|---|---|---|
| CLEAN | 0 | 89.09 | 0.1205 | 0.0000 | 0.4760 | 0.4744 | 0.4761 | 0.5240 | 0.5256 | 0.5239 |
| EV_NOISE | 0.05 | 88.92 | 0.1238 | 0.0032 | 0.4757 | 0.4742 | 0.4758 | 0.5243 | 0.5258 | 0.5242 |
| | 0.10 | 88.80 | 0.1249 | 0.0062 | 0.4756 | 0.4742 | 0.4758 | 0.5244 | 0.5258 | 0.5242 |
| | 0.20 | 88.68 | 0.1252 | 0.0104 | 0.4757 | 0.4743 | 0.4758 | 0.5243 | 0.5257 | 0.5242 |
| | 0.30 | 88.64 | 0.1238 | 0.0133 | 0.4757 | 0.4743 | 0.4758 | 0.5243 | 0.5257 | 0.5242 |
| | 0.50 | 88.35 | 0.1216 | 0.0198 | 0.4758 | 0.4745 | 0.4759 | 0.5242 | 0.5255 | 0.5241 |
| IMG_NOISE | 0.05 | 88.57 | 0.1261 | 0.0258 | 0.4764 | 0.4749 | 0.4766 | 0.5236 | 0.5251 | 0.5234 |
| | 0.10 | 87.90 | 0.1295 | 0.0548 | 0.4769 | 0.4755 | 0.4771 | 0.5231 | 0.5245 | 0.5229 |
| | 0.20 | 87.05 | 0.1290 | 0.1066 | 0.4779 | 0.4766 | 0.4780 | 0.5221 | 0.5234 | 0.5220 |
| | 0.30 | 86.57 | 0.1238 | 0.1445 | 0.4788 | 0.4776 | 0.4790 | 0.5212 | 0.5224 | 0.5210 |
| | 0.50 | 85.74 | 0.1097 | 0.2021 | 0.4808 | 0.4798 | 0.4809 | 0.5192 | 0.5202 | 0.5191 |

Table 11: Fusion metrics on the UCF-Crime dataset under varying noise settings. AUC is reported as a percentage.

| Noise Type | Masked Level | AP (%) | Brier | KL | $\Delta w_e$ | $\Delta w_e^{\mathrm{ab}}$ | $\Delta w_e^{\mathrm{n}}$ | $\Delta w_x$ | $\Delta w_x^{\mathrm{ab}}$ | $\Delta w_x^{\mathrm{n}}$ |
|---|---|---|---|---|---|---|---|---|---|---|
| CLEAN | 0 | 88.26 | 0.0735 | 0.0000 | 0.4661 | 0.4623 | 0.4672 | 0.5339 | 0.5377 | 0.5328 |
| EV_NOISE | 0.05 | 88.12 | 0.0741 | 0.0020 | 0.4661 | 0.4623 | 0.4672 | 0.5339 | 0.5377 | 0.5328 |
| | 0.10 | 88.00 | 0.0747 | 0.0040 | 0.4661 | 0.4623 | 0.4671 | 0.5339 | 0.5377 | 0.5329 |
| | 0.20 | 87.74 | 0.0761 | 0.0083 | 0.4660 | 0.4624 | 0.4671 | 0.5340 | 0.5376 | 0.5329 |
| | 0.30 | 87.51 | 0.0769 | 0.0121 | 0.4660 | 0.4624 | 0.4670 | 0.5340 | 0.5376 | 0.5330 |
| | 0.50 | 87.13 | 0.0792 | 0.0194 | 0.4658 | 0.4623 | 0.4668 | 0.5342 | 0.5377 | 0.5332 |
| IMG_NOISE | 0.05 | 87.95 | 0.0794 | 0.0267 | 0.4665 | 0.4628 | 0.4676 | 0.5335 | 0.5372 | 0.5324 |
| | 0.10 | 87.64 | 0.0852 | 0.0546 | 0.4670 | 0.4633 | 0.4680 | 0.5330 | 0.5367 | 0.5320 |
| | 0.20 | 86.60 | 0.0986 | 0.1162 | 0.4679 | 0.4644 | 0.4689 | 0.5321 | 0.5356 | 0.5311 |
| | 0.30 | 85.86 | 0.1109 | 0.1788 | 0.4687 | 0.4655 | 0.4697 | 0.5313 | 0.5345 | 0.5303 |
| | 0.50 | 84.23 | 0.1366 | 0.3082 | 0.4705 | 0.4677 | 0.4714 | 0.5295 | 0.5323 | 0.5286 |

Table 12: Fusion metrics on the XD-Violence dataset under varying noise settings.

| Noise Type | Masked Level | AUC (%) | Brier | KL | $\Delta w_e$ | $\Delta w_e^{\mathrm{ab}}$ | $\Delta w_e^{\mathrm{n}}$ | $\Delta w_x$ | $\Delta w_x^{\mathrm{ab}}$ | $\Delta w_x^{\mathrm{n}}$ |
|---|---|---|---|---|---|---|---|---|---|---|
| CLEAN | 0 | 98.17 | 0.0402 | 0.0000 | 0.4718 | 0.4633 | 0.4723 | 0.5282 | 0.5367 | 0.5277 |
| EV_NOISE | 0.05 | 98.13 | 0.0401 | 0.0006 | 0.4720 | 0.4634 | 0.4725 | 0.5280 | 0.5366 | 0.5275 |
| | 0.10 | 98.11 | 0.0398 | 0.0022 | 0.4722 | 0.4635 | 0.4727 | 0.5278 | 0.5365 | 0.5273 |
| | 0.20 | 98.07 | 0.0399 | 0.0038 | 0.4726 | 0.4638 | 0.4731 | 0.5274 | 0.5362 | 0.5269 |
| | 0.30 | 98.06 | 0.0393 | 0.0053 | 0.4729 | 0.4639 | 0.4734 | 0.5271 | 0.5361 | 0.5266 |
| | 0.50 | 98.06 | 0.0383 | 0.0082 | 0.4734 | 0.4641 | 0.4739 | 0.5266 | 0.5359 | 0.5261 |
| IMG_NOISE | 0.05 | 97.66 | 0.0430 | 0.0162 | 0.4726 | 0.4647 | 0.4731 | 0.5274 | 0.5353 | 0.5269 |
| | 0.10 | 97.54 | 0.0434 | 0.0308 | 0.4733 | 0.4651 | 0.4738 | 0.5267 | 0.5349 | 0.5262 |
| | 0.20 | 96.29 | 0.0477 | 0.0758 | 0.4750 | 0.4678 | 0.4754 | 0.5250 | 0.5322 | 0.5246 |
| | 0.30 | 95.09 | 0.0520 | 0.1084 | 0.4764 | 0.4696 | 0.4768 | 0.5236 | 0.5304 | 0.5232 |
| | 0.50 | 90.72 | 0.0611 | 0.2180 | 0.4801 | 0.4754 | 0.4803 | 0.5199 | 0.5246 | 0.5197 |

Table 13: Fusion metrics on the ShanghaiTech dataset under varying noise settings.

Across all four datasets, our method consistently demonstrates robust uncertainty-guided fusion behavior. When degradation is applied to the motion-cue modality ($z_e$), performance remains stable—AUC or AP typically drops by less than 1%, and uncertainty weights show minimal change. This suggests that the model is not overly sensitive to motion-cue corruption and maintains reliable fusion under partial degradation.

| Noise Type | Masked Level | AUC (%) | Brier | KL | $\Delta w_e$ | $\Delta w_e^{ab}$ | $\Delta w_e^{n}$ | $\Delta w_x$ | $\Delta w_x^{ab}$ | $\Delta w_x^{n}$ |
|---|---|---|---|---|---|---|---|---|---|---|
| CLEAN | 0 | 92.39 | 0.1119 | 0.0000 | 0.4807 | 0.4786 | 0.4814 | 0.5193 | 0.5214 | 0.5186 |
| EV_NOISE | 0.05 | 92.27 | 0.1126 | 0.0024 | 0.4808 | 0.4787 | 0.4815 | 0.5192 | 0.5213 | 0.5185 |
| | 0.10 | 92.19 | 0.1129 | 0.0048 | 0.4810 | 0.4788 | 0.4816 | 0.5190 | 0.5212 | 0.5184 |
| | 0.20 | 91.94 | 0.1149 | 0.0088 | 0.4812 | 0.4791 | 0.4819 | 0.5188 | 0.5209 | 0.5181 |
| | 0.30 | 91.89 | 0.1153 | 0.0116 | 0.4814 | 0.4791 | 0.4821 | 0.5186 | 0.5209 | 0.5179 |
| | 0.50 | 91.86 | 0.1168 | 0.0158 | 0.4817 | 0.4794 | 0.4824 | 0.5183 | 0.5206 | 0.5176 |
| IMG_NOISE | 0.05 | 92.27 | 0.1117 | 0.0066 | 0.4811 | 0.4790 | 0.4817 | 0.5189 | 0.5210 | 0.5183 |
| | 0.10 | 92.10 | 0.1118 | 0.0125 | 0.4816 | 0.4795 | 0.4822 | 0.5184 | 0.5205 | 0.5178 |
| | 0.20 | 91.86 | 0.1111 | 0.0220 | 0.4824 | 0.4804 | 0.4830 | 0.5176 | 0.5196 | 0.5170 |
| | 0.30 | 91.63 | 0.1116 | 0.0303 | 0.4832 | 0.4813 | 0.4838 | 0.5168 | 0.5187 | 0.5162 |
| | 0.50 | 91.52 | 0.1106 | 0.0425 | 0.4848 | 0.4827 | 0.4854 | 0.5152 | 0.5173 | 0.5146 |

Table 14: Fusion metrics on the MSAD dataset under varying noise settings.

In contrast, masking the image modality ($z_x$) produces more pronounced effects, especially on appearance-dependent datasets such as UCF-Crime and XD-Violence. At the highest masking level, UCF-Crime experiences a 2.35% drop in AUC, and XD-Violence shows nearly a 5% drop in AP, accompanied by KL divergence increases up to 0.3082. Uncertainty weights reflect this asymmetry: $w_x$ consistently decreases while $w_e$ increases, indicating that the model dynamically downweights unreliable image features and reallocates confidence toward the motion-cue modality.

On datasets with higher modality redundancy—such as ShanghaiTech and MSAD—both performance and uncertainty remain relatively stable under perturbation. ShanghaiTech maintains an AUC above 98% under motion-cue noise and above 90% under severe image masking, while MSAD exhibits only minor fluctuations across all metrics. This confirms that the model can maintain balanced modality fusion when both modalities offer sufficient information.

While the average change in uncertainty weights ($\Delta w_m$) is numerically small—typically below 1%—this is largely due to aggregation over all vector indices and time steps. A finer-grained analysis reveals that individual latent dimensions can shift by as much as 30% under value-level masking, demonstrating substantial feature-wise modulation. Moreover, uncertainty reallocation is more pronounced in abnormal segments compared to normal ones, indicating that the model adapts more sensitively in semantically critical regions. Finally, increases in Brier score under corruption reflect growing misalignment between predicted probabilities and ground truth labels, particularly under image degradation, further confirming that the model's confidence dynamically adjusts in response to input quality.

## L   VISUALIZATION OF ANOMALY DETECTION WITH UNCERTAINTY

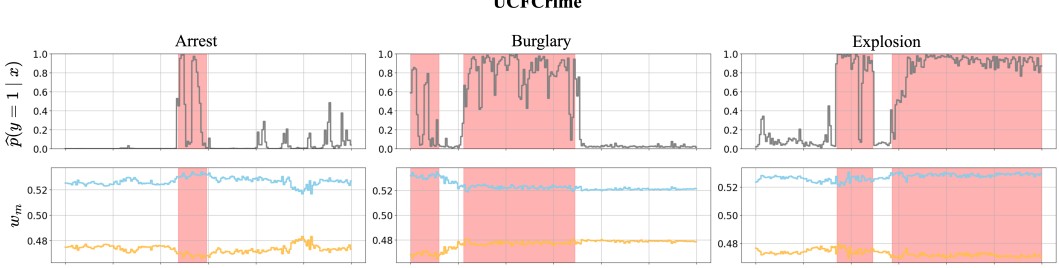

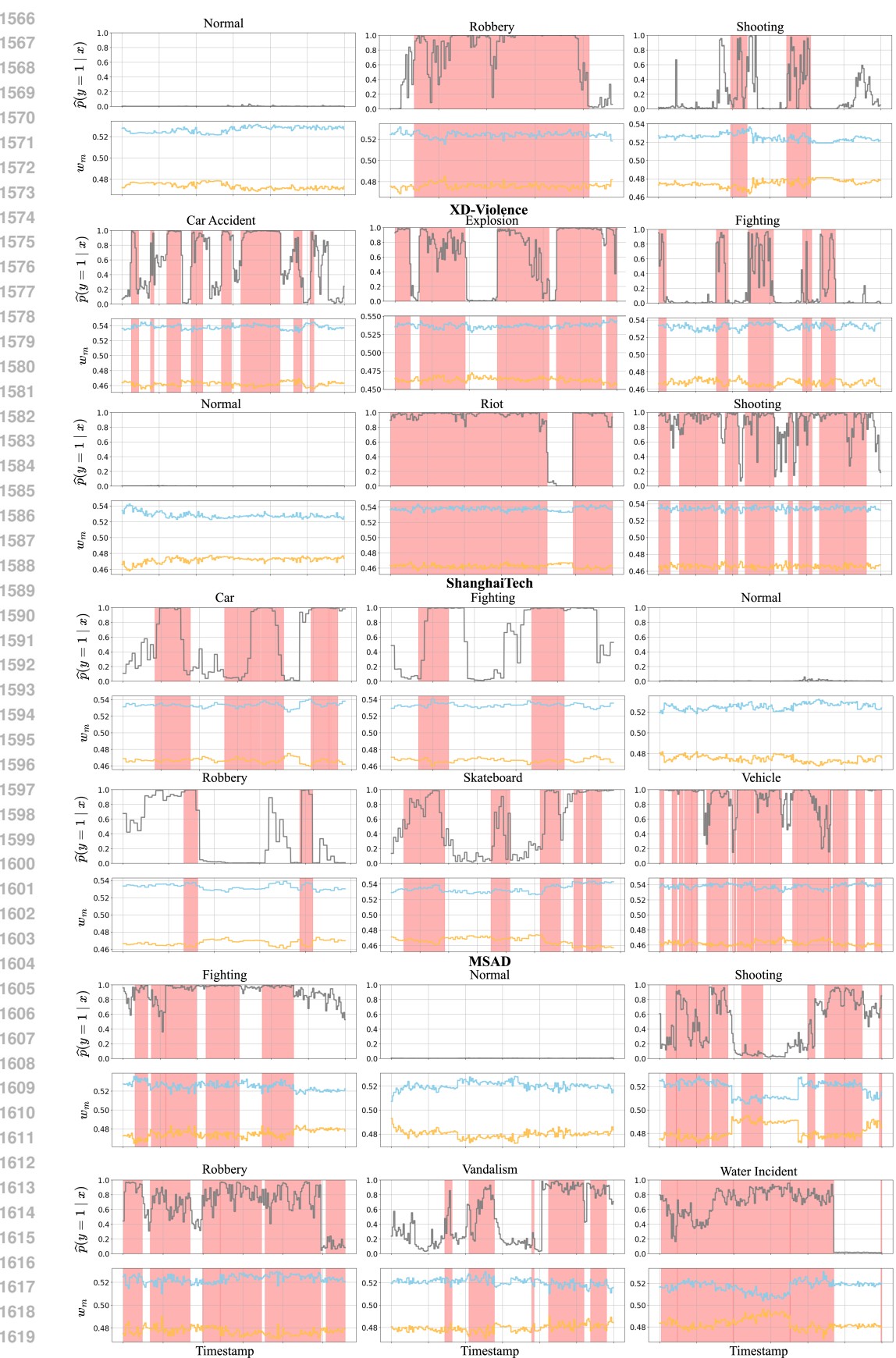

