# OpenReview forum: "Uncertainty-Weighted Fusion of RGB and Synthetic Motion Cues for Video Anomaly Detection"
_ICLR.cc/2026/Conference — ICLR 2026 Conference Withdrawn Submission_

### Official Review · Reviewer_YJcF · 2025-10-28

**Soundness:** 2
**Presentation:** 2
**Contribution:** 3
**Rating:** 4
**Confidence:** 4

**Summary:**

This work proposes an uncertainty-weighted fusion of RGB and motion cues for improving the performance of video anomaly detection.

They model the variance of feature representations to remove cross-modal noise.

Experiments are conducted on several benchmarks and show improved performance, including the newest MSAD dataset.

**Strengths:**

+ Overall the paper is well organised, with theoretical insights, nice visualisations and plots.

+ The experimental evaluations and comparisons are solid and clearly show the effectiveness of the proposed method.

+ The authors explore a diverse representation of experimental visualisations to show the insights of the proposed method.

**Weaknesses:**

**Major**

- Related work section could be improved by adding some recent works in discussions and analysis, such as newest works from 2025. It would be better to outline how the proposed method differs from existing ones.

- It would be better to have a notation section detailing the maths symbols and operations used in the paper, e.g., what are scalars, vectors, and matrices, etc. In the current version, it seems hard for me to tell what variables are vectors etc.

- Regarding evaluations, it would be more interesting to see a complete evaluation on MSAD dataset regarding scenario-level and anomaly-type-level evaluations. These evaluations would strengthen and increase the impact of this work.

- Figure font sizes, table font sizes should be consistent in the whole paper. Some figure font sizes are too small to read, such as Fig. 3. Also this figure is a bit hard to understand, it would be better if the authors could provide some extra explanations.

**Minor**

- It is suggested to reduce the heavy use of “—”, as that seems like machine generated symbols. Also it is suggested not to heavily use bullet points (lists of items), as those look a bit unprofessional in academic writing.

**Questions:**

Please refer to weaknesses.

I would be supportive if the authors could address the major concerns, particularly regarding related works and evaluations.

---

### Official Review · Reviewer_vPjf · 2025-10-30

**Soundness:** 2
**Presentation:** 2
**Contribution:** 2
**Rating:** 4
**Confidence:** 4

**Summary:**

The paper introduces an uncertainty-weighted multimodal fusion framework for video anomaly detection. The approach combines RGB inputs with synthetic motion cues generated from standard video frames. It models feature noise using a Student-t distribution and applies a Laplace approximation to compute inverse-variance weights for fusion. Experiments are conducted on several public datasets, showing moderate performance gains compared to existing methods. The work focuses on integrating uncertainty modeling into the fusion process but follows established multimodal fusion principles.

**Strengths:**

1. The paper addresses a relevant setting where event cameras are unavailable, using synthetic motion cues derived from RGB frames.
2. The method applies a Student-t noise model with Laplace approximation for uncertainty-based fusion, providing a probabilistic interpretation of feature weighting.
3. The experimental section includes quantitative comparisons across multiple datasets with clear tabular presentation.

**Weaknesses:**

1. While the method is solid, it primarily combines existing Bayesian and uncertainty-based fusion principles. The overall framework remains within established multimodal fusion paradigms without introducing a fundamentally new theoretical insight.
2. No direct comparison among different noise assumptions (Gaussian, Laplace, Student-t). No cross-dataset or noisy-environment generalization experiments are provided.
3. The generation process and quantitative characteristics of the synthetic motion cue are not fully explained or contrasted with real event streams.

**Questions:**

1. While the model achieves good results on specific datasets, how does it perform on other types of videos such as low light, complex backgrounds, or events from different domains? Has the model been evaluated for cross-domain or cross-environment applications, and how does it avoid overfitting?
2. To what extent can synthetic motion cues capture the microsecond-level temporal resolution and extreme fast motion that real event sensors such as DVS can capture? Can this synthetic data handle the demands of event streams in extreme scenarios such as fast-moving objects or instantaneous motion?

---

### Official Review · Reviewer_5KAt · 2025-10-31

**Soundness:** 3
**Presentation:** 3
**Contribution:** 2
**Rating:** 4
**Confidence:** 5

**Summary:**

This paper tackles weakly supervised video anomaly detection (VAD) and proposes EF-VAD, a probabilistic fusion framework that combines RGB frame features with synthetic motion cues obtained from simple inter-frame differencing and temporal aggregation.  Each modality produces a mean and a variance, and the fusion is performed using Inverse- variance (precision) weighting, where uncertain features contribute less. To handle noise in motion cues, the authors use a Student-t likelihood and approximate it with a Laplace-based Gaussian for efficient computation.  Extensive experiments are conducted on UCF-Crime, XD-Violence, ShanghaiTech, and MSAD datasets.

**Strengths:**

- Clear motivation and relevance:

 The paper addresses an important challenge—how to combine motion cues and RGB features effectively without one dominating the other. The uncertainty-based weighting is a meaningful step in that direction.


- Practical and deployable approach:

 The use of synthetic motion cues derived from RGB differences provides a low-cost alternative to real event sensors, making the method more practical for large-scale or real-world applications.

**Weaknesses:**

- Performance not at par with current state-of-the-art:

While IEF-VAD achieves stable and competitive results, it does not surpass existing top-performing methods on key benchmarks. For instance, on UCF-Crime, PI-VAD reports 90.33 AUC vs. 88.67 in this paper and on XD-Violence, PEMIL reports 88.21 AP vs. 87.63 in this paper. This indicates that although uncertainty-aware fusion is effective, the improvement is incremental rather than groundbreaking, especially compared to advanced transformer-based or prompt-based frameworks.


- Limited novelty beyond fusion formulation:

Conceptually, the overall framework remains similar to other multimodal fusion systems such as those based on cross-attention or gating. The main novelty lies in the type of weighting (inverse-variance) and the use of Student-t/Laplace modeling, which are well-established ideas in Bayesian inference and sensor fusion literature. The contribution therefore feels more like an adaptation of existing probabilistic ideas to VAD rather than a fundamentally new anomaly modeling approach.

- Simplistic design of synthetic motion cues:

The so-called synthetic events are created using a very simple process, frame differencing + thresholding + temporal averaging. While this simplicity can be practical, it raises several questions:

-- Why not compare with optical flow magnitudes, temporal gradients with learnable thresholds, or lightweight motion encoders that could capture richer dynamics?

-- Without such baselines, it is unclear whether the observed improvements come from the fusion mechanism or the choice of motion representation.

This design decision limits the representational novelty of the proposed framework.


- Overemphasis on “abrupt motion cues”:

The paper assumes that “abrupt or transient motions” are the main indicators of anomalies. This assumption holds for violent or accident-type anomalies but not for subtle human-centric cases like shoplifting or loitering, where motion is slow and context-dependent. Hence, focusing only on sharp motion cues may limit the system’s ability to capture fine-grained temporal patterns. The authors could have discussed this limitation or integrated models that handle both sharp and subtle motion changes.

- Questionable choice of backbone encoders:

If motion is central to the approach, using a CLIP image encoder (static in nature) for frame-level feature extraction seems suboptimal. Spatio-temporal encoders such as I3D, VideoSwin, or VideoMAE-v2 would better capture motion patterns. The authors should clarify why choosing CLIP over other spatio-temporal encoders.

- Missing computational efficiency and scalability analysis:

Although the paper claims the method is lightweight and practical, there are no quantitative results to support this claim.  Important details such as, FPS, FLOPs and parameter count should be reported and compared against existing multimodal fusion methods (e.g., cross-attention, gating). This analysis is especially critical since the iterative refinement module likely adds extra computation.

- No cross-dataset generalization results:

The paper trains and evaluates models separately for each dataset, but there is no experiment showing cross-dataset transferability. Because the framework learns dataset-specific motion statistics, it might overfit to the type of anomalies in each dataset. Testing a model trained on UCF-Crime (violent anomalies) on ShanghaiTech (less violent, more object-centric anomalies) would provide valuable insights into generalization and robustness.

- Fusion is not always beneficial:

Figure 2 in the paper shows that for certain classes (e.g., Fighting or Assault), the fusion model performs worse than motion cues alone.  This suggests that the proposed uncertainty weighting sometimes fails to exploit complementary information between modalities. The authors should analyze why this happens—whether due to variance estimation bias, refinement instability, or over-reliance on RGB features.

- Missing qualitative visualizations:

The paper would benefit from visual examples of Uncertainty (σ²) maps, Spatial or temporal weight distributions, and Before–after refinement feature changes. Such visualizations would greatly help the reader understand what the model learns and how uncertainty is being allocated between modalities.


- Missing critical evaluation metrics:

Since the goal is improved anomaly discrimination, the authors should report AUC for anomaly segments only (AUC_A), as done in UMIL and PI-VAD. This would clarify whether performance gains come from better anomaly detection or simply more accurate normal classification. Current AUC metrics, which mix normal and abnormal samples, obscure this distinction. Further, the author should go for Event based evaluation such as (temporal IOU) to verify the robustness of the method.

**Questions:**

Justify the novelty for cross modal fusion.

 Why overemphasis on “abrupt motion cues”?

Why is CLIP the choice of backbone encoders?

Why is Fusion not always beneficial?

Why chosing a shallow qualitative analysis?

---

### Official Review · Reviewer_6TWw · 2025-11-03

**Soundness:** 3
**Presentation:** 2
**Contribution:** 2
**Rating:** 4
**Confidence:** 5

**Summary:**

The paper proposes IEF-VAD (Uncertainty-Weighted Image-Event Fusion for Video Anomaly Detection), a framework that integrates RGB frames with synthetic motion cues derived from inter-frame differencing. These motion cues serve as event-like proxies that highlight transient and localized dynamics absent in static images. The method employs a Bayesian uncertainty-weighted fusion strategy, assigning inverse-variance weights to balance modality contributions under noise and feature imbalance. By modeling heavy-tailed uncertainty with a Student’s t distribution and refining the fused latent state iteratively.

**Strengths:**

The proposed method is technically solid. Its performance is promising, showing strong potential and effectiveness in addressing video anomaly detection.

**Weaknesses:**

The motivation for introducing synthetic motion cues is not clearly explained, and this design introduces additional computational overhead. Please refer to the related questions for specific concerns.

**Questions:**

1. Could the authors clarify the architecture of the event encoder? Specifically, on what dataset is it pre-trained, and is it frozen or fine-tuned during training?

2. Figure 3 is difficult to interpret and follow. Could the authors improve its clarity or provide more detailed explanations to enhance reader understanding?

3. How does the proposed method’s uncertainty estimation mechanism differ from those used in ECU[1] and LEC-VAD[2]? What are the advantages of the proposed approach?

4. The paper mentions “dynamic motion cues”, but their exact definition and motivation are not clearly explained. Could the authors clarify what these cues represent and why they are introduced?

5. Has the paper conducted ablation studies on different feature combinations—for example, using only CLIP-based features or removing the motion-cue branch? This would help verify the contribution of each component.

6. There are several missing punctuation marks and citation formatting errors throughout the manuscript. The authors are encouraged to carefully proofread and correct these issues before resubmission.


[1] Exploiting Completeness and Uncertainty of Pseudo Labels for Weakly Supervised Video Anomaly Detection

[2] Learning Event Completeness for Weakly Supervised Video Anomaly Detection

---

### Note · Authors · 2025-11-12

I have read and agree with the venue's withdrawal policy on behalf of myself and my co-authors.